# Molecular-level insight into photocatalytic $CO_2$ reduction with $H_2O$ over Au nanoparticles by interband transitions

Wenchao Shangguan[1], Qing Liu[1], Ying Wang [2 ✉], Ning Sun[1], Yu Liu [3], Rui Zhao[3], Yingxuan Li [1 ✉], Chuanyi Wang[1] & Jincai Zhao [4]

Achieving $CO_2$ reduction with $H_2O$ on metal photocatalysts and understanding the corresponding mechanisms at the molecular level are challenging. Herein, we report that quantum-sized Au nanoparticles can photocatalytically reduce $CO_2$ to CO with the help of $H_2O$ by electron-hole pairs mainly originating from interband transitions. Notably, the Au photocatalyst shows a CO production rate of 4.73 mmol $g^{-1}\,h^{-1}$ (~100% selectivity), ~2.5 times the rate during $CO_2$ reduction with $H_2$ under the same experimental conditions, under low-intensity irradiation at 420 nm. Theoretical and experimental studies reveal that the increased activity is induced by surface Au–O species formed from $H_2O$ decomposition, which synchronously optimizes the rate-determining steps in the $CO_2$ reduction and $H_2O$ oxidation reactions, lowers the energy barriers for the *CO desorption and *OOH formation, and facilitates CO and $O_2$ production. Our findings provide an in-depth mechanistic understanding for designing active metal photocatalysts for efficient $CO_2$ reduction with $H_2O$.

[1] School of Environmental Science and Engineering, Shaanxi University of Science and Technology, Xi'an 710021, China. [2] State Key Laboratory of Rare Earth Resource Utilization, Changchun Institute of Applied Chemistry, Chinese Academy of Sciences, Changchun 130022, China. [3] Engineering Research Center of Advanced Functional Material Manufacturing of Ministry of Education, School of Chemical Engineering, Zhengzhou University, Zhengzhou 450001, China. [4] Key Laboratory of Photochemistry, CAS Research/Education Center for Excellence in Molecular Sciences, Institute of Chemistry, Chinese Academy of Sciences, Beijing 100190, P. R. China. ✉email: ywang_2012@ciac.ac.cn; liyingxuan@sust.edu.cn

Coupling $CO_2$ reduction and $H_2O$ oxidation over photo-catalysts using solar energy is an appealing strategy to alleviate the greenhouse effect and simultaneously produce value-added chemicals or fuels[1,2]. In recent decades, photocatalytic studies have almost exclusively focused on semiconductors, which provide an opportunity for reducing $CO_2$ with $H_2O$ at room temperature[3,4]. However, the reaction rates of semiconductor photocatalysts are usually proportional to the square root of light intensity, thereby limiting their efficiency in practical applications, in which relatively high light intensity is required[5].

Other than semiconductor photocatalysts, some plasmonic (such as Ag, Au, and Cu)[6–12] or nonplasmonic (such as Ni and Ru)[13,14] metal nanoparticles (NPs) have been recently reported to show photocatalytic activities for $CO_2$ reduction. Under illumination, a collective oscillation of free electrons (conduction band electrons) is sometimes induced in metal NPs, a phenomenon defined as localized surface plasmon resonance (LSPR). Moreover, valence electrons in metal NPs can be excited to a much higher energy level through interband transitions[15]. Compared with the photoinduced electron-hole pairs in semiconductors, charge carriers produced in metal NPs usually possess sufficient energy for activating chemical bonds[6,16]. In addition, metal photocatalysts also have the ability to combine light and thermal energies to activate inert molecules in catalytic reactions[15]. Notably, a linear increase in reaction rates with increasing reaction temperature and light intensity was reported for metal photocatalysts, which sharply contrasts with semiconductor photocatalysts[17]. These intriguing properties make metal photocatalysts effective candidates for overcoming the severe challenges to photocatalytic $CO_2$ reduction[18].

However, for photocatalytic $CO_2$ conversion on metal NPs, strong reductants (such as $CH_4$, $H_2$, isopropyl alcohol, and triethanolamine) are generally required[7,19–21]. For application purposes, using inexpensive and abundant $H_2O$ as an electron donor should be the direction and goal of photocatalytic $CO_2$ reduction[22]. In this reaction, a series of complex and consecutive multi charge transfer processes followed by simultaneous $CO_2$ reduction and $H_2O$ oxidation reactions at two different sites should occur under illumination[3]. The hot carriers generated from the LSPR effect of metal photocatalysts always have a much shorter lifetime than those from semiconductors[23], which leads to difficulty in accomplishing $CO_2$ reduction with $H_2O$. To solve this problem, combining plasmonic metal nanostructures with semiconductors (such as $TiO_2$) has been a widely used method by which the lifetime of hot charge carriers is prolonged[24]. In this situation, the metal NPs work as photosensitizers via the LSPR effect, and their low interfacial charge transfer efficiency greatly limits the application of these composite photocatalysts[25].

In contrast to the LSPR effect, hot electron-hole pairs generated from the interband transitions of metal NPs always possess higher redox potentials and longer lifetimes, which are more conducive to driving the $H_2O$ oxidation and $CO_2$ reduction reactions[26]. However, to the best of our knowledge, photocatalytic $CO_2$ reduction with $H_2O$ has been rarely realized on metal-based photocatalysts by interband transitions. Moreover, molecular-level insights into the fundamentals of photocatalytic reactions on metal photocatalysts are still challenging to obtain and lacking, greatly restricting the rational design of high-efficiency metal photocatalysts. Gold (Au) NPs have received increasing attention for the photocatalytic conversion of $CO_2$ into C1 or C2 hydrocarbons by the LSPR effect, which is mainly performed in the presence of $H_2$ or sacrificial agents[6,19–21,27]. Recently, it has been proven that the interband absorption of Au NPs can dominate plasmon absorption with decreasing particle size[28]. Therefore, quantum-sized Au NPs with unique interband excitations under visible light irradiation may provide unprecedented possibilities for driving the photocatalytic conversion of $CO_2$ with $H_2O$.

In this work, quantum-sized Au NPs with a diameter of ~4 nm, which can be used as efficient photocatalysts for reducing $CO_2$ to CO with $H_2O$ by interband transitions, are synthesized by a one-pot reduction approach. Molecular insight into the photocatalytic reaction mechanism reveals that the $CO_2$ reduction and $H_2O$ oxidation reactions on Au followed the $CO_2 \rightarrow *COOH \rightarrow *CO \rightarrow CO(g)$ and $H_2O \rightarrow *OH \rightarrow *O^- \rightarrow *O \rightarrow *OOH \rightarrow O_2$ routes, respectively. Notably, the surface Au–O species formed from decomposing $H_2O$ are proven to be able to lower the activation energy of the $CO_2$ reduction reaction ($CO_2RR$) and improve the evolution of CO and $O_2$ on the Au surface.

## Results

**Characterizations of the Au NPs.** The ~4 nm Au NPs were fabricated by reducing $HAuCl_4 \cdot xH_2O$ with $NaBH_4$ at room temperature. The representative transmission electron microscopy (TEM) image in Fig. 1a shows that the as-prepared Au NPs exhibit quasispherical morphology. As shown in the enlarged image (Supplementary Fig. 1), accidental aggregation of the Au NPs is observed, which is understandable considering that the surfactant-free synthesis strategy was used in the present study. The high-resolution TEM (HRTEM) image in Fig. 1b shows that the lattice spacing of the nanoparticle is 0.235 nm, consistent with the spacing of the (111) planes of metallic Au. As shown in Fig. 1c, the Au NPs display a narrow size distribution in the range of $4.0 \pm 0.5$ nm. X-ray photoelectron spectroscopy (XPS) was employed to study the valence states of the obtained samples (Fig. 1d). The binding energies centered at 83.6 and 87.2 eV prove the metallic character of the Au NPs[29].

As revealed by the UV–Vis absorption spectrum in Fig. 1e, the Au NPs exhibit absorption in the wavelength range of 320–800 nm. A small absorption peak from 485 to 650 nm with maximum absorption at 520 nm is ascribed to the LSPR effect due to intraband absorption, which occurs by the excitation of free conduction electrons near the Fermi surface from 6sp-hybridized atomic orbitals of Au (Fig. 1f). In addition to absorption from the LSPR effect, a dominant absorption band with a tail that can reach 800 nm is observed in Fig. 1e, which can be attributed to the 5d–6sp interband transitions of Au NPs (Fig. 1f)[26,30]. According to theoretical calculations[31], the interband absorption of Au NPs is derived from two types of excitations. As shown in Fig. 1f, one type of excitation occurs by the transition of electrons from the top of the 5d band to states just above $E_F$ (Fermi level) in the 6sp band with a threshold at 2.4 eV ($\lambda < 517$ nm, occurs near the L-point in the Brillouin zone), and the other is from the 5d band to unoccupied states in the 6sp band above $E_F$ with a tail at 1.8 eV ($\lambda < 689$ nm, occurs near the X-point in the Brillouin zone). Considering that minor discrepancies between the theoretical prediction and experimental observation should be acceptable, the intraband and interband absorption regions of Au are marked with different colors in Fig. 1e. As shown in Fig. 1e, the absorption of Au NPs in the range of 320–800 nm is attributed to the overlap of the LSPR intraband absorption and the interband absorption.

**Photocatalytic performance.** The photocatalytic activity of the as-obtained Au NPs towards $CO_2$ reduction with $H_2O$ was evaluated in a steel reactor with a quartz window on top, with temperature-controlled by a heating element. To evaluate the contributions of the LSPR and interband transitions to photocatalysis, monochromatic light-emitting diode (LED) light sources at wavelengths in the visible region (from 365 to 620 nm) were used. The intensities of the LED light sources are summarized in

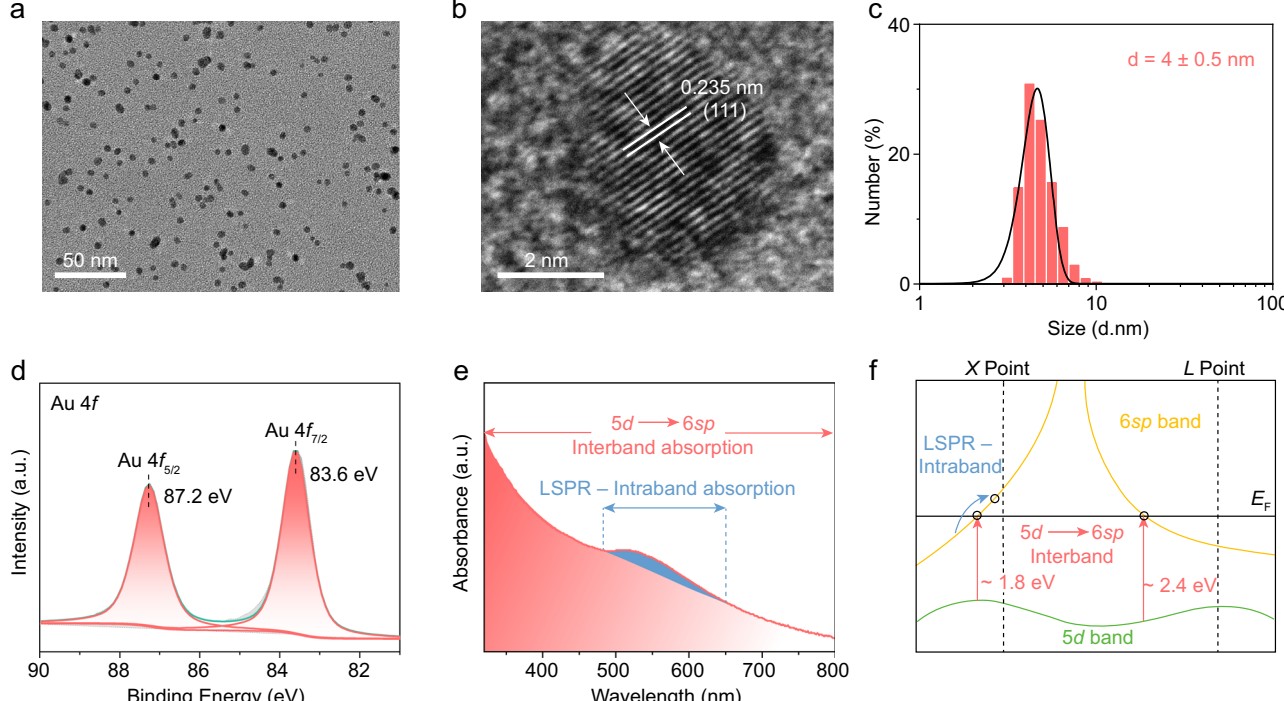

**Fig. 1 Materials characterization. a** Typical TEM image, **b** HRTEM image, and **c** Particle size distribution of Au NPs. **d** Au 4 *f* XPS spectra of as-prepared Au NPs. **e** UV–Vis absorption spectrum of the Au NPs with interband and intraband absorption. **f** Schematic diagram of interband and intraband (LSPR) transitions.

Supplementary Table 1. Figure 2a shows the photocatalytic CO production activities at 200 °C under 420 nm LED light irradiation. The Au NPs show CO evolution up to 4.73 mmol g$^{-1}$ h$^{-1}$ accompanied by 1.98 mmol g$^{-1}$ h$^{-1}$ O$_2$ evolution (based on a 4 h reaction), which are much higher than most of the semiconductor-based photocatalysts, even with the low-intensity LED light used in our study (Supplementary Table 2).

Figure 2b shows the apparent quantum efficiency (AQE) of the Au NPs for CO production under light with different wavelengths, which decreases with increasing wavelength in the range of 365 to 620 nm, except for a slight increase at 520 nm. The enhanced AQE at 520 nm should be ascribed to the improved light absorption due to the LSPR effect of the Au NPs[32]. Notably, the AQE is fairly low under irradiation ranging from 450 to 620 nm, which is unlike previous observations in which the action spectra were well-matched with the LSPR spectra of Au NPs[32]. This observation provides clear evidence that the CO evolution reaction was not primarily driven by the LSPR effect of Au[26,30]. In contrast, the main parts of the AQEs at 365 nm and 420 nm match well with the absorption induced by the interband transitions (Fig. 2b), proving that the photocatalytic CO$_2$-to-CO conversion on Au NPs should be mainly attributed to the hot electrons from interband transitions[26,30]. Furthermore, the absorption cross-section of the Au NPs in the range of 320–485 nm increases at shorter wavelengths (Supplementary Fig. 2), which is in agreement with the AQE trend in Fig. 2b. This result further confirms that the photocatalytic CO$_2$RR activity is correlated with the 5*d*–6*sp* interband transitions of Au[33].

To gain insight into the dynamics of the photogenerated carriers, the time-resolved photoluminescence (TRPL) decay spectrum of Au NPs was obtained (Supplementary Fig. 3). The carrier lifetime of the Au sample was calculated to be approximately 0.2 ns, which is close to that of semiconductor photocatalysts and is long enough to drive photocatalytic reactions[34]. Interestingly, this lifetime is also much longer than

that produced with the LSPR effect (approximately 30 fs)[23], which is beneficial for the transfer of photogenerated charge carriers and thereby leads to improved photocatalytic activity on Au NPs. The TRPL result proves the superiority of the charge carriers induced by the interband transition of Au for photocatalysis.

As a control experiment, in the absence of catalyst or CO$_2$, there no products were detected even after 4 h of reaction (Fig. 2c). To confirm the role of light and temperature in the catalytic CO$_2$ reduction reactions, control experiments without light or heat input were also performed. As shown in Fig. 2c, negligible products were observed when CO$_2$ reduction was performed at 200 °C on Au NPs without light, while CO production was only 0.181 mmol g$^{-1}$ h$^{-1}$ under continuous illumination with 420 nm LED light without additional heating. The above results reflect that CO evolution over Au is caused by CO$_2$ reduction under contributions from both light and thermal energy.

As we know, thermal splitting of CO$_2$ or H$_2$O is only reportedly achieved on metal oxide-based catalysts by using oxygen vacancies as the reaction intermediate, which usually requires significantly high temperatures above 1000 °C[35]. Although thermal conversion of CO$_2$ on Au can be achieved at approximately 200 °C, H$_2$ must be used as a reducing agent in this reaction[5]. Therefore, thermocatalytic reduction of CO$_2$ with H$_2$O can hardly be achieved on Au at 200 °C (Fig. 2c). Furthermore, the photothermal effect on catalytic CO$_2$ reduction on Au NPs can be neglected because 420 nm LED light with low intensity (73 mW cm$^{-1}$, even lower than the solar intensity) was used in the present study. Under 420 nm LED light irradiation, the temperature of the sample can only increase from ambient temperature (24 °C) to 33 °C. Based on the above considerations, we can speculate that CO$_2$-to-CO conversion on Au NPs should be reduced by the photogenerated electrons.

However, a linear dependence of the photocatalytic CO production rate on the reaction temperature at constant light

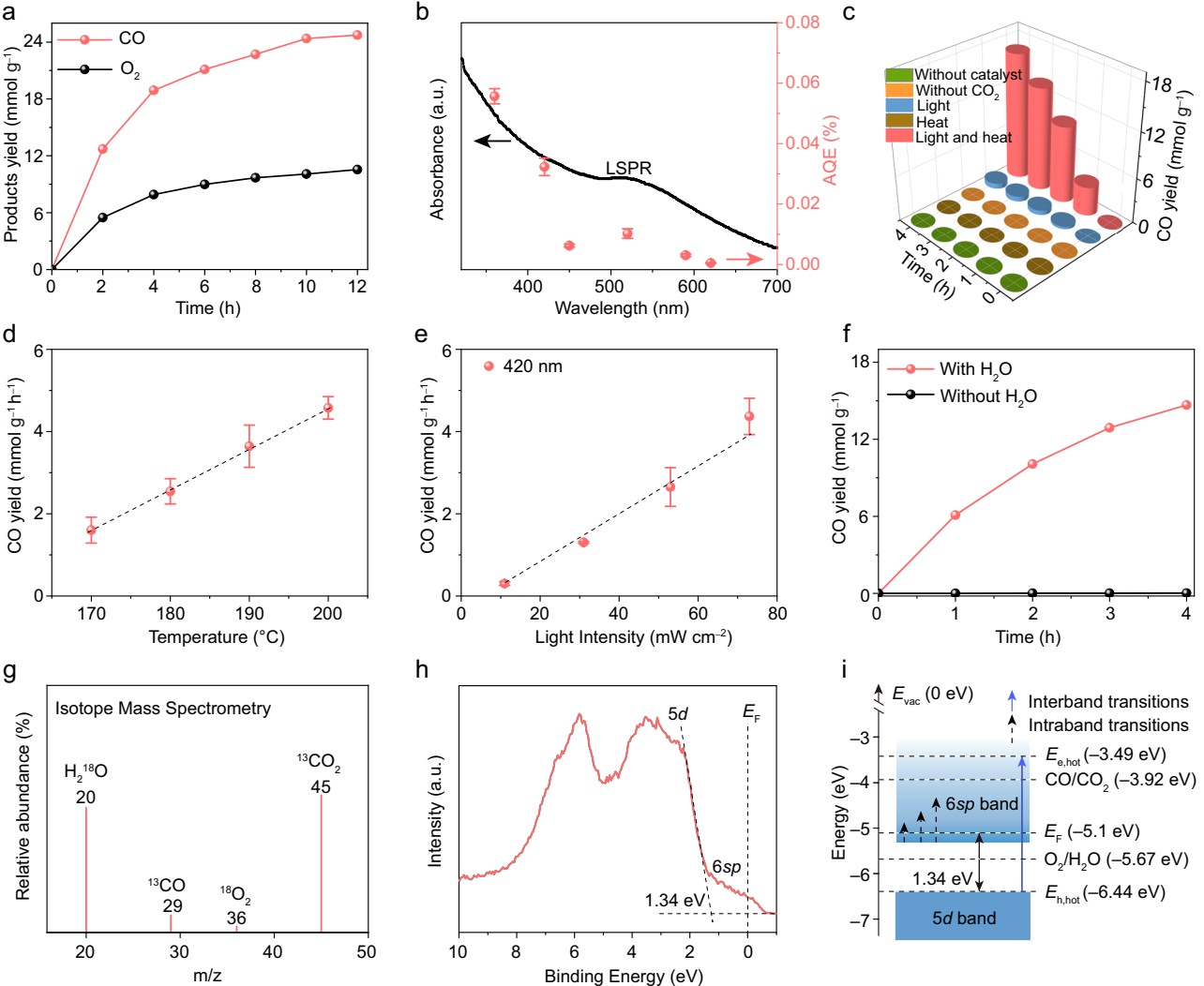

**Fig. 2 CO₂RR performance. a** CO and O₂ productions over Au NPs with 420 LED light irradiation at 200 °C. **b** Wavelength-dependent AQEs of the Au photocatalyst ($n = 3$, error bars: standard deviation). **c** Control experiments with different reaction conditions. **d** The reaction temperatures and **e** light intensity-dependent CO production rate on Au NPs under 420 nm LED light irradiation ($n = 3$, error bars: standard deviation). **f** Photocatalytic CO₂RR performance with (red curve) or without (black curve) H₂O on Au NPs. **g** Mass spectrum from GC-MS analysis of the ¹³CO and ¹⁸O₂ generated from photocatalytic ¹³CO₂RR with H₂¹⁸O over Au NPs. **h** Valence-band XPS spectrum of Au. **i** Schematic illustration for the electronic band structure of the ~4 nm Au NPs.

intensity is shown in Fig. 2d, which indicates that the heat input can play a role in improving the photocatalytic CO₂RR activity of Au. This phenomenon is possibly induced by the improved populations of the adsorbate in excited vibrational states at higher temperatures[17]. In the future, the influence of the reaction temperature on photocatalytic CO₂ conversion on Au needs to be systematically studied. Moreover, a nearly linear increase in CO production with increasing light intensity is shown in Fig. 2e, which is a characteristic of reactions on metal photocatalysts[17]. This result confirms that the CO₂RR on Au is in fact a light-driven process. The results in Fig. 2d and Fig. 2e make the Au NPs practical for increasing the reaction rate by increasing the temperature and light intensity, indicating that the Au NPs in our work can efficiently couple thermal and light energy to drive chemical transformations, which is in sharp contrast to semiconductor photocatalysts[25].

To reveal the role of H₂O in the photocatalytic CO₂ reduction process over Au, desorption of absorbed H₂O from the surface of Au was carried out by calcining the sample at 450 °C for 1 h in Ar gas, and the complete desorption of H₂O was confirmed by FT-IR

spectra (Supplementary Fig. 4). After that, photocatalytic CO₂ reduction without H₂O was conducted at 200 °C under 420 nm LED illumination, and hardly any CO evolution was detected (Fig. 2f, black curve). However, when H₂O was added to the reactor, the photocatalytic CO evolution rate on Au reached 3.7 mmol g⁻¹ h⁻¹ (Fig. 2f, red curve), which was similar to the rate in Fig. 2a, confirming that H₂O is essential for the CO₂RR. Furthermore, ¹³CO₂ and H₂¹⁸O isotopic labeling experiments confirm that CO production was derived from the reduction of CO₂ along with the oxidation of H₂O into O₂ (Fig. 2g).

To prove that the Au sample has a suitable electronic band structure for the CO₂ reduction and H₂O oxidation reactions, the valence-band XPS spectra of the sample were examined. As shown in Fig. 2h, the zero point of the energy scale corresponds to the $E_F$ position. Two broad peaks at 3 eV and 6 eV result from the splitting of the $5d_{5/2}$ and $5d_{3/2}$ levels by spin-orbit interaction, while the $5d$ band edge of Au is approximately 1.34 eV[36]. Moreover, an electronic state induced by the $6sp$ electrons of Au tailing above $E_F$ reflects the metallic character of the Au NPs[28]. Based on the result in Fig. 2h, the electronic band structure of the

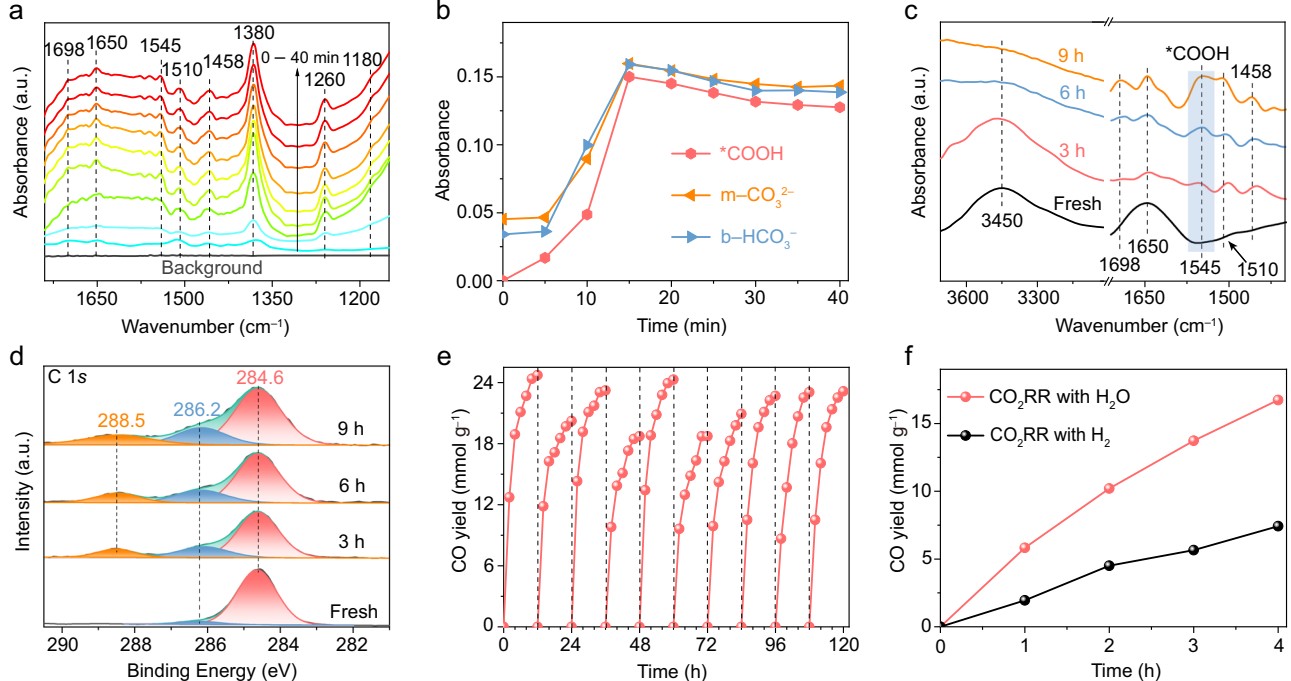

**Fig. 3 Characterization of the formed species during CO₂RR and the enhanced CO₂RR performance in H₂O compared with that in H₂. a** In situ FT-IR spectra of the photocatalytic CO₂ reduction process occurring with $H_2O$ on Au. FT-IR spectra were recorded every 5 min from bottom to top. **b** Absorbance of the intermediate species that formed on Au during the photocatalytic CO₂RR. **c** FT-IR and **d** high-resolution C 1 *s* XPS spectra at photocatalytic reaction times of 0 (fresh), 3, 6, and 9 h. **e** Stability test for Au NPs under 420 nm illumination for CO₂RR at 200 °C. **f** Photocatalytic CO₂RR performance on Au NPs with $H_2O$ or $H_2$ under 420 nm light illumination at 200 °C. The volume ratio of $H_2$ to $CO_2$ is 4:1 in the gas mixture.

Au NPs and the charge transfer processes involved in the photocatalytic CO₂ reduction and $H_2O$ oxidation reactions are schematically shown in Fig. 2i. The value of hot electron energy ($E_{e,hot}$) is −3.49 eV when the Au NPs are illuminated by 420 nm LED light, which is more positive than the reduction potential of $CO/CO_2$ (−3.92 eV vs. vacuum level)[37]. The value of the corresponding hot hole energy ($E_{h,hot}$) is determined to be −6.44 eV, which is more negative than the oxidation potential of $O_2/H_2O$ (−5.67 eV vs. vacuum level)[33]. The details for calculating $E_{e,hot}$ and $E_{h,hot}$ are shown in the Method part. These results indicate that the Au NPs exhibit a suitable electronic band structure to achieve CO₂ reduction and $H_2O$ oxidation simultaneously by interband transitions.

It is well known that selectivity is also a fundamental issue for the CO₂RR with $H_2O$ due to the competing $H_2$ evolution reaction (HER). In the present study, no $H_2$ production on Au was detected during photocatalytic CO₂ reduction with $H_2O$ (Supplementary Fig. 5). The results in Fig. 3c-d prove that the *COOH intermediate can be strongly adsorbed on the active sites of the Au NPs (discussed below), which is a possible reason for the HER being suppressed since the active sites are occupied by *COOH. On the other hand, the absorbed *COOH can increase the reaction barrier for the HER on Au NPs[38], which can also contribute to the suppression of the HER. In addition, the weak affinity between Au and H might be another factor responsible for suppressing the HER on Au[39]. For the above three reasons, selective CO₂ reduction with the inhibition of the side HER is achieved on the Au photocatalyst in the present study.

**Fourier transform infrared spectroscopy studies**. To elucidate the catalytic mechanism, in situ Fourier transform infrared

(in situ FT-IR) spectroscopy was conducted to detect the surface-adsorbed species on Au during the photocatalytic process. As shown in Fig. 3a, a broad peak centered at 1650 cm⁻¹ corresponding to the surface-adsorbed $H_2O$ appears with prolonged reaction time, suggesting that $H_2O$ was adsorbed on the Au NPs. As the reaction proceeded, a new peak at 1698 cm⁻¹ that can be assigned to the vibration of carboxylate ($CO_2^-$) was observed, which represents the chemical adsorption state of $CO_2$ on Au[40]. The interaction between the $CO_2$ and $H_2O$ (and/or surface oxygen atom) coadsorbed on the Au surface leads to the mono-dentate carbonate (m-$CO_3^{2-}$) peaks at 1510 cm⁻¹ and 1380 cm⁻¹ and bidentate bicarbonate (b-$HCO_3^{2-}$) peak at 1458 cm⁻¹ [40,41]. Moreover, two new peaks at 1545 and 1180 cm⁻¹ that can be ascribed to the *COOH group (the asterisk denotes the surface adsorption site)[40] appear with increasing reaction time, demonstrating that *COOH is a key intermediate of the $CO_2$-to-CO conversion process. Moreover, the changes in the peak intensities of the critical intermediates with reaction time are shown in Fig. 3b, which increase with the extension of the reaction time from 0 to 15 min. After that, the peak intensities are slightly reduced with the reaction time being prolonged from 15 to 30 min and are nearly stable after 30 min of reaction. The changes in peak intensities may be understood by considering the cooperative effect between the $H_2O$ and $CO_2$ adsorption process and the photocatalytic conversion process of the intermediate species during the reduction of $CO_2$ on Au NPs.

As shown in Fig. 2a, the $CO_2$-to-CO conversion rate is greatly reduced after 3 h of photocatalytic reaction. To study the reasons for this phenomenon, FT-IR spectra over long time ranges were tested (Fig. 3c). As shown in Fig. 3c, the intensities of the peaks belonging to *COOH increase gradually as the reaction time proceeds to 9 h, suggesting that an accumulation of *COOH on

the Au surface occurs during the $CO_2RR$ process. XPS analysis was also conducted to further verify the gradual accumulation of *COOH species on the Au surface (Fig. 3d). For the Au catalyst after photoreactions, there is an increase in the C–O peak intensity at 286.2 eV and a simultaneous emergence of a peak at 288.5 eV, which can be attributed to the adsorbed intermediate *COOH on the Au surface during the photocatalytic $CO_2$ reduction reaction[42]. This is basically consistent with the FT-IR result in Fig. 3c. *COOH accumulation might induce the deactivation of the Au photocatalysts observed in Fig. 2a by blocking active sites. To desorb the *COOH species, the Au sample after 12 h of $CO_2RR$ was outgassed under vacuum for 12 h, and the elimination of *COOH was verified by the FT-IR (Supplementary Fig. 6) and XPS (Supplementary Fig. 7) spectra. Then, the decarboxylated sample was reused to convert $CO_2$ and $H_2O$ under the same reaction conditions as those in Fig. 2a, and the CO production activity recovered immediately. As shown in Fig. 3e, the Au NPs remained stable for at least 10 cycles over the entire 120 h reaction, indicating that the Au catalyst possesses good stability.

## Discussion

It is well known that the $CO_2RR$s on metal photocatalysts were mainly performed in $H_2$ gas in the previous reports[5]. Therefore, photocatalytic $CO_2$ conversion on Au was also assessed by substituting $H_2O$ with $H_2$ under the same conditions. Sample dehydration before the $CO_2RR$ was carried out to eliminate the influence of adsorbed $H_2O$. With the help of $H_2$, the CO yield on Au is 1.86 mmol $g^{-1} h^{-1}$ (Fig. 3f), which is less than 1/2 of that formed in Fig. 2a. Gas chromatography-mass spectrometry (GC–MS) demonstrates that CO is the only reduced product in the $CO_2RR$ with $H_2$ (Supplementary Fig. 8). This result clearly reveals that the unique quantum-sized Au NPs exhibit superiority in photocatalytic $CO_2$ reduction with $H_2O$. Therefore, the present work opens an exciting pathway to efficiently drive $CO_2$ reduction with $H_2O$ on metal NPs by interband transitions, which is essential for fundamental research and applications. To understand this counterintuitive phenomenon, the role of $H_2O$ in $CO_2RR$ is systematically studied based on experiments and theoretical calculations in the following.

To study the reaction pathway for $H_2O$ during the $CO_2RR$, $^1H$ solid-state nuclear magnetic resonance ($^1H$ ssNMR) spectra of the Au NPs before and after the photocatalytic reaction were obtained. As shown in Fig. 4a, in the spectra of the fresh Au NPs, two different types of H species can be clearly observed. The broad peak with a chemical shift of 6.36 ppm is caused by highly immobilized $H_2O$ molecules with restricted mobility near the surface region[43]. The sharp peak at 3.43 ppm can be attributed to the weakly adsorbed $H_2O$ on the outer surface of the Au[44,45]. Furthermore, surface $H_2O$ species on Au can be also confirmed by thermogravimetric analysis (TGA). As shown in Fig. 4b, there are two major stages in weight loss from room temperature to 450 °C in the derivative thermogravimetry (DTG) curve. The first weight loss of ~2.3% occurred from the initial temperature to 258 °C, which is attributed to the sustained release of weakly adsorbed $H_2O$ (physisorbed water) from the surface of Au[46]. The second weight loss of ~1.2% in the temperature range of 258-450 °C can be attributed to the removal of highly immobilized water from the surface of Au[47]. This is well consistent with the $^1H$ ssNMR results.

When the catalyst was subjected to photocatalytic $CO_2RR$ for 3 h, the peak at 6.36 ppm disappeared (Fig. 4a), suggesting that the immobilized $H_2O$ molecules were decomposed under illumination. Simultaneously, a dominant resonance centered at 3.96 ppm emerged in the red curve, which is ascribed to the

overlapping signals between OH and free $H_2O$, indicating the formation of an $AuOH-H_2O$ species on the Au surface by hydrogen bonds between AuOH groups and free $H_2O$[44]. In addition, the peak centered at 1.47 ppm in the red curve can be related to the isolated *OH species (without hydrogen bonds) formed on the Au surface[44]. These results are very similar to the observations of the production of $SiOH–H_2O$ and isolated SiOH species on mesoporous silica NPs[44].

Based on the $^1H$ ssNMR spectra, it can be concluded that molecular $H_2O$ was dissociated into *OH species under photocatalytic reaction conditions and that partial *OH species were not bound to $H_2O$. This is consistent with the in situ FT-IR characterizations in Fig. 4c, in which a stretching band (at 3740 $cm^{-1}$) attributed to isolated *OH species on the Au surface was observed[48]. And a characteristic band at 1260 $cm^{-1}$ that can be regarded as the vibration of the *OOH group appeared after the $CO_2RR$ (Fig. 3a), which is regarded as an important intermediate for $O_2$ production in the $H_2O$ oxidation process[49].

To study the species formed on Au during the photocatalytic process, quasi in situ XPS measurements were carried out. As shown in Fig. 4d, prior to irradiation, the high-resolution XPS spectrum of Au $4f$ exhibits two peaks at 83.3 eV and 86.9 eV, which represent the Au $4f_{7/2}$ and Au $4f_{5/2}$ states of metallic Au ($Au^0$)[50], respectively. Upon irradiation at 200 °C, the peaks corresponding to the oxide state of Au ($Au^+$) emerged at 84.8 eV and 88.4 eV[50], which demonstrates that the Au–O species formed by $H_2O$ decomposition on Au during the reaction process. As shown in Fig. 4d, the concentration of the Au–O species gradually increased as the reaction time was prolonged from 1 h to 3 h, which coincides with the production of Au–O species in the photocatalytic $CO_2RR$ on Au.

To further confirm the formation of Au oxide species, electrochemical linear scan voltammetry (LSV) was performed on the Au samples before and after the photocatalytic reaction. As shown in Fig. 4e, compared to the fresh Au sample, a peak at approximately 0.8 V vs. the reversible hydrogen electrode (RHE) can be clearly observed for the Au sample after 3 h of photocatalytic reaction. As proven by the cyclic voltammetry (CV) curve of Au NPs (Supplementary Fig. 9), the onset potential during the back scan is not sufficient to oxidize Au. Therefore, the peak at 0.8 V can be attributed to the reduction of the generated Au oxide species during the photocatalytic process[51]. In addition, Raman spectroscopy was also used to characterize the Au–O species formed on the Au surface. As shown in Fig. 4f, a characteristic peak at 480 $cm^{-1}$ appears during the photocatalytic $CO_2RR$ process, which can be attributed to the stretching vibrations of Au–O species[52]. Both the electrochemistry and Raman characterizations confirm that Au–O species formed on the surface of Au NPs during the photocatalytic $CO_2RR$, which is consistent with the XPS result in Fig. 4d.

Furthermore, the high-resolution O $1s$ XPS spectra are conducted to distinguish the oxygen species on Au surface. Supplementary Fig. 10 shows the changes of XPS spectra before and after reaction under 420 nm LED light irradiation. Two peaks at 531.7 eV and 533.1 eV in both samples are observed, which are attributed to the hydroxyl groups and surface adsorbed $H_2O$ molecules, respectively[53]. While the O $1s$ spectrum of the Au sample after 3 h reaction shows two additional shoulder peaks at 530.0 eV and 535.0 eV, the former can be attributed to the Au–O species[54], which is consistent with the XPS results of Au $4f$. The latter peak at 535.0 eV can be attributed to chemisorbed oxygen species from COOH or/and $H_2O$[55]. As discussed above through $^1H$ ssNMR, FT-IR, XPS, electrochemistry, and Raman analyses, we can confirm that Au–O species were formed on the Au surface in the photocatalytic $CO_2RR$ process by decomposing $H_2O$. Thus, we speculate that the formed Au–O species might play a crucial

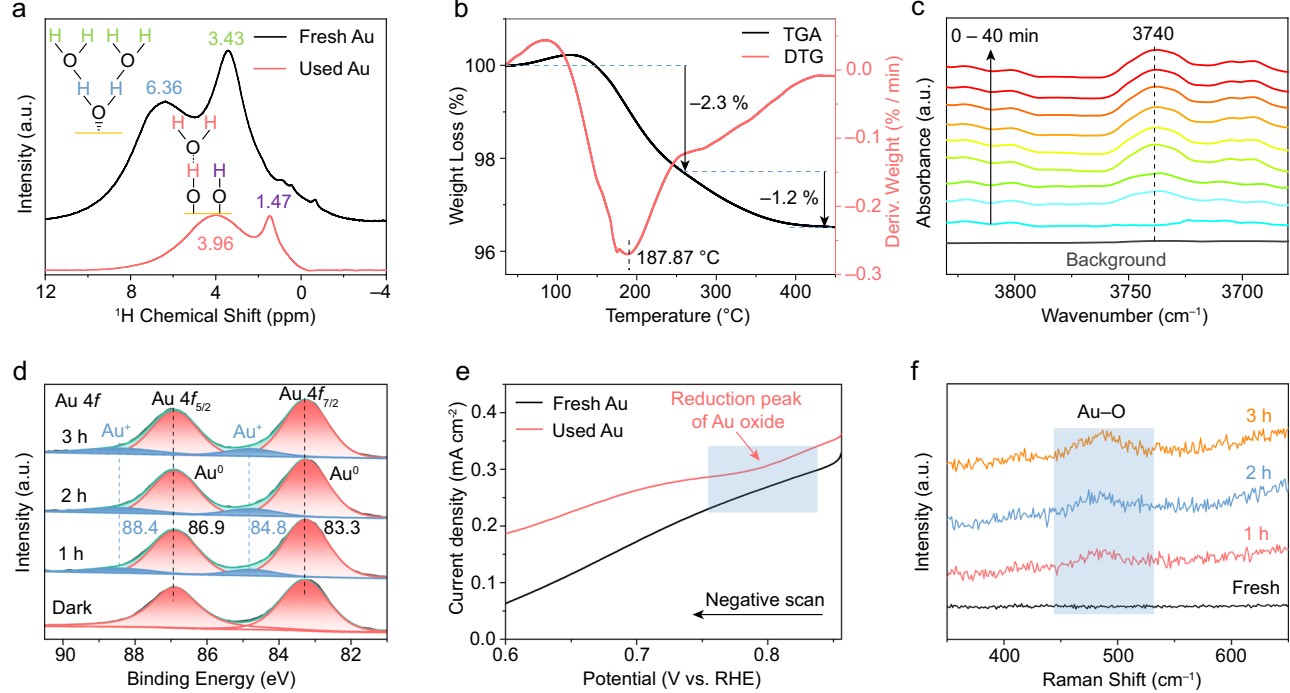

**Fig. 4 Characterization of surface Au–O species formed by $H_2O$ decomposition. a** [1]H ssMAS-NMR spectra of Au samples before (black curve) and after $CO_2RR$ (3 h reaction, red curve). **b** TGA-DTG curves of the fresh Au photocatalyst. **c** In situ FT-IR spectra of the photocatalytic $CO_2$ reduction with $H_2O$ on Au. FT-IR spectra were recorded every 5 min from bottom to top. **d** Quasi in situ XPS (Au 4 *f*) of Au NPs at photocatalytic reaction times of 0 (fresh), 1, 2, and 3 h. **e** LSV curves of fresh and used (3 h reaction) Au catalyst. **f** Raman spectra of Au NPs at photocatalytic reaction times of 0 (fresh), 1, 2, and 3 h.

role in $CO_2RR$ on Au. Meanwhile, the above results prove that the *COOH, *OH, and *OOH species are the main intermediates in photocatalytic $CO_2$ reduction with $H_2O$.

Based on the above experimental results, we employed density functional theory (DFT) calculations based on an electrochemical method to examine the possible $CO_2RR$ mechanisms occurring on the surface of Au. The experimental results above have proven that the photocatalytic $CO_2$ reduction and $H_2O$ oxidation processes over Au NPs undergo pathways (Eqs. (1–9)) that are the same as the electrochemical reduction pathways of $CO_2$ in $H_2O$[56]. Furthermore, the Gibbs free energy diagram estimated by the electrochemical simulation method has been widely used in investigating the mechanism of photocatalytic $CO_2RR$ considering that the thermodynamic properties of the chemical reactions should not be changed by light stimulation[9,41,57]. Based on the above considerations, the electrochemical simulation method might be suitable to investigate the photocatalytic $CO_2RR$ mechanism on Au in terms of thermodynamics.

We first simulated the energetic barriers for the $H_2O$ dissociation reaction on Au (111) surface without photoinduced charges. As shown in Fig.5a, an extremely high energy barrier (1.35 eV) with an endothermic process (1.09 eV) is observed, indicating that it is difficult to produce $H^+$ on the Au surface by direct $H_2O$ decomposition without the participation of photoinduced charges. Considering that $CO_2$ reduction via the proton-coupled electron transfer process is favourable[21], it was reasonable that the $CO_2RR$ on the Au surface was negligible in $H_2O$ without the presence of light since no available $H^+$ and $e^-$ pairs were generated. However, according to the previous theoretical calculations, the energy barrier for $H_2O$ decomposition would be significantly reduced if the crossing of excited states is considered[58,59]. Therefore, both the presence of $H_2O$ and light illumination are critical for the $CO_2RR$ on Au, which is consistent with our experimental observations in Fig. 2c. Furthermore, according to the above [1]H ssNMR, FT-IR, and XPS analyses, light

irradiation is helpful for dissociating $H_2O$ into *OH and H*, then goes through the *O and *OOH intermediates and eventually forms $O_2$. In this process, O bound to the Au surface inevitably formed under light irradiation. Therefore, the improved $CO_2RR$ activity on Au with $H_2O$ in Fig. 3f most likely resulted from the effect of surface Au–O species. As a result, a model of the Au surface with preadsorbed O was constructed to simulate the $CO_2RR$ with $H_2O$.

To explore the effect of Au–O species on the $CO_2RR$ and oxygen evolution reaction (OER) activities, we calculated the free energy diagrams for CO and $O_2$ evolution on Au with (O-preadsorbed Au) and without (clean Au) surface Au–O species. As shown in Fig. 5b, the electronic configuration regulated by the Au–O species can modify the adsorption of intermediates formed during the $CO_2RR$. Furthermore, the required free energy of the rate-determining step (the first protonation process of $CO_2$ to form *COOH) on O-preadsorbed Au (0.59 eV) is lower than that (the last step of *CO desorption) on clean Au (0.68 eV), implying the higher activity of the O-preadsorbed Au surface in the $CO_2RR$. Figure 5b also shows that the affinity between CO and Au is weakened by the presence of the Au–O species. As shown in Fig. 5b, the reaction energy for *CO desorption is reduced from 0.68 eV to 0.33 eV in the presence of surface Au–O species, indicating that the adsorbed CO can be more easily eliminated on Au if $H_2O$ is used for $CO_2$ reduction. This is also helpful for enhancing the $CO_2$-to-CO conversion activity on Au. As shown in Fig. 5c, the thermodynamic barrier for OER (determined by the rate-determining step of *O + $H_2O$→*OOH + $H^+$ + $e^-$) is greatly reduced on Au in the presence of surface Au–O species (1.59 eV) compared with a clean Au surface (2.64 eV), demonstrating that the preadsorbed O can act as an effective structure to boost the OER activity by weakening the adsorption of the formed intermediates in the $H_2O$ oxidation process.

To understand the mechanism by which the Au–O species enhance the above $CO_2RR$ and OER performances, we further

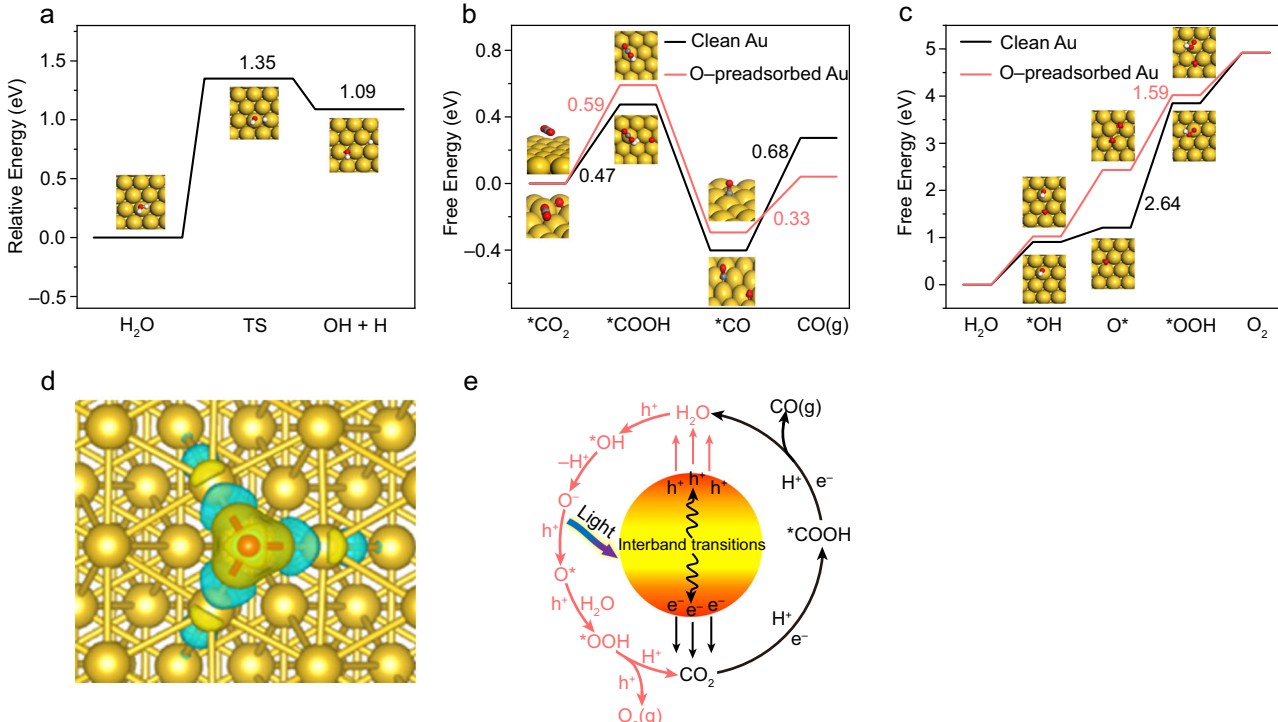

**Fig. 5 DFT simulation results and catalytic mechanism. a** The potential energy curves for $H_2O$ dissociation on Au without light irradiation. Free energy diagram for **b** $CO_2RR$ and **c** $H_2O$ oxidation on O pre-adsorbed Au and clean Au. The golden, red, and white balls represent Au, O, and H atoms, respectively. **d** Calculated charge density differences for O pre-adsorbed Au surface. The charge accumulation and depletion are colored in yellow and cyan. **e** Reaction pathways for the photoreduction $CO_2RR$ with $H_2O$ into CO and $O_2$ on the Au NPs.

examined the electronic structure of the Au centers in the Au–O species using Bader charge analysis. As shown in Fig. 5d, apparent charge transfer occurs on the O-preadsorbed Au surface by transferring the electrons from the $d$ orbital of Au to O. As a result, the d electrons present on the Au atoms involved in adsorbing the intermediates (*COOH/*CO/*OH/O*/*OOH species) of the $CO_2RR$ and OER processes are reduced, verifying the superiority of Au–O for the enhancement of CO and $O_2$ production. The relationship of catalytic activity, adsorption energy and $d$ electrons obtained here is consistent with our previous studies[60]. Overall, based on the DFT calculation results, we can confirm that the generated Au–O species play a critical role in synchronously improving the $CO_2RR$ and OER activities on Au, which is consistent with experimental results in Fig. 3f.

According to experimental results and DFT calculations, a possible reaction mechanism for the photocatalytic reduction of $CO_2$ with $H_2O$ is proposed as follows. Under illumination, hot electron-hole pairs are generated on the Au NPs through interband transitions (Fig. 5e). The photogenerated holes can efficiently dissociate $H_2O$ into $O_2$ based on the following equations:

$$h^+ + Au + H_2O \rightarrow Au-OH + H^+ \quad (1)$$

$$Au-OH \rightarrow Au-O^- + H^+ \quad (2)$$

$$h^+ + Au-O^- \rightarrow AuO \quad (3)$$

$$h^+ + AuO + H_2O \rightarrow AuOOH + H^+ \quad (4)$$

$$h^+ + AuOOH \rightarrow O_2 + H^+ \quad (5)$$

Unlike the $H^+$ generated from direct $H_2O$ decomposition, the $H^+$ released from Au–OH (Au–OH→Au–O⁻ + $H^+$) can possibly participate in photocatalytic $CO_2$ reduction by the proton-coupled electron transfer process. In this process, negatively

charged Au–O⁻ species are left on the Au surface[61]. The formation of Au–O⁻ species was confirmed by zeta potential measurements. Compared with the zeta potential of the fresh Au sample (–2.71 mV), the Au sample after 9 h of photocatalytic reaction possesses a much more negative potential (–14.3 mV), consistent with the formation of Au–O⁻ species on the surface of Au in the photocatalytic process[61]. Moreover, the adsorbed $CO_2$ on the Au NP surface reacts with photogenerated e⁻ and $H^+$ to yield CO based on the following equations:

$$CO_2 + Au \rightarrow Au-CO_2 \quad (6)$$

$$Au-CO_2 + H^+ + e^- \rightarrow Au-COOH \quad (7)$$

$$Au-COOH + H^+ + e^- \rightarrow Au-CO + H_2O \quad (8)$$

$$Au-CO \rightarrow CO(g) + Au \quad (9)$$

## Methods

**Catalyst preparation**. Au NPs were routinely prepared by a one-step reduction method. In detail, 3 mL ice-cooled $NaBH_4$ (98%, Aladdin) solution (0.1 M) was rapidly added into a 100 mL aqueous solution containing 0.25 mM $HAuCl_4·xH_2O$ (99%, Adamas) under vigorous magnetic stirring. In this process, the color of the solution changed from yellow to wine red, which matched the formation of Au NPs. After stirring for 30 min, the obtained solution was aged for 24 h to completely hydrolyze the unreacted $NaBH_4$. Then, the aged solution containing Au NPs was dialyzed in deionized water for a week to remove unnecessary ions, in which the water was exchanged for fresh water each day.

**Materials characterization**. XPS and valence band XPS characterization was conducted with a ThermoFisher ESCALAB 250Xi instrument equipped with an excitation source of monochromatic aluminium, and the C1s peak at 284.6 eV can be regarded as the reference for the calibration binding energy. TEM and HRTEM images were acquired on an FEI Tecnai G2 F20 microscope working at 200 kV. The absorbance spectra of the samples were collected on a Shimadzu UV-2600 UV−Vis spectrophotometer, in which the dialyzed Au colloidal solution was dropped onto a

sample plate with $BaSO_4$ as the background, and the diffuse reflectance spectra were measured after drying. The absorption cross-section of 4 nm Au NPs was calculated using the method in a previous report[33]. Raman microscopy was performed using the emission of a 633 nm laser. The TRPL spectrum was obtained on an Edinburgh FS-5 spectrometer with an excitation wavelength of 340 nm. FT-IR was carried out on a Bruker VERTEX 70 V. TGA was measured on a TGA 5500 instrument with a heating rate of 10 °C min$^{-1}$ under nitrogen flow. $^1$H ssNMR spectra were obtained on an Agilent 600 DD2 spectrometer at room temperature. The resonance frequency and the spinning rate used in this test were 600.19 MHz and 15 kHz, respectively. Zeta potential measurements were performed on a NAMO-ZS Zetasizer (Malvern, UK). Isotopic distributions of the gas products were analysed on a gas chromatograph-mass spectrometer (GCMS-QP2010 plus, Shimadzu).

**Electrochemical measurements**. Electrochemical measurements were performed in 0.1 M $H_2SO_4$ electrolyte at room temperature with an electrochemical workstation (CHI 760E, CH Instruments, Inc.) with a standard three-electrode system. The prepared sample, Pt wire, and a saturated Ag/AgCl electrode were used as working, counter, and reference electrodes, respectively. The working electrode was prepared by attaching Au samples onto a 1 × 1 cm fluorine-doped tin oxide glass substrate through dilute Nafion solution (1%). CV and LSV curves were acquired from 0.6 V to 0 V vs. Ag/AgCl at a scan rate of 20 mV/s. The potentials vs. Ag/AgCl were converted to the RHE scale by the following equation:

$$E_{RHE} = E_{Ag/AgCl} + 0.197 + 0.059 \times pH \qquad (10)$$

**In situ FT-IR measurements**. In situ FT-IR measurements were carried out on a Bruker Vertex 70 v spectrometer equipped with a liquid nitrogen-cooled HgCdTe detector. All spectra were obtained with a resolution of 2 cm$^{-1}$ and an accumulation time of 32 s. A photo of the instrument used for in situ FT-IR measurements is shown in Supplementary Fig. 11. Prior to the measurement, a vacuum pump was used to evacuate all the gases from the reaction chamber for 1 h. Then, $CO_2$ and $H_2O$ vapour, as reaction gases, were introduced into the reaction chamber through the gas valve. The temperature of the reaction cell was raised to 200 °C with a heating rate of 5 °C min$^{-1}$. After that, the background spectrum in the presence of the sample was collected. Next, the FT-IR spectra were recorded as a function of time, in which 420 nm LED light was used to illuminate the sample by a top quartz window.

**Quasi in situ XPS measurements**. Quasi in situ XPS spectra were acquired with a customized SPECS UHV system equipped with a SPECS PHOIBOS 150 analyser, a SPECS XR50 X-ray source, and a SPECS 1D DLD detector. The basic pressure of the analysis chamber was less than $3 \times 10^{-10}$ mbar. The Au catalyst used for the XPS measurement was pressed onto stainless steel mesh. The $CO_2$ photoreduction process was conducted in a load-lock chamber. After loading the catalyst, the reaction chamber was first treated with 5 mbar $CO_2$ and $H_2O$ vapour and then heated to 200 °C for the photocatalytic reaction. In this process, 420 nm LED light was used to illuminate the sample. The distance between the light source and the sample was approximately 13.5 cm. After the photocatalytic reactions proceeded for different reaction times (1 h, 2 h, or 3 h), the load-lock chamber was pumped down to high vacuum ($<10^{-9}$ mbar). Then, the sample was transferred to the analysis chamber through the sample transfer axis for XPS measurements without exposing the sample to air.

**Photocatalytic activity tests**. Photocatalytic $CO_2$ conversion with $H_2O$ was carried out in a custom-built stainless-steel reaction chamber (450 mL) with a quartz window on top for light irradiation. The temperature of the reaction system was controlled by a heating element. Before the photocatalytic reaction test, 3 mL of the prepared Au colloidal solution (~100 mg L$^{-1}$) was transferred into a quartz container and then dried in a vacuum freeze dryer overnight. The obtained Au photocatalyst was evenly distributed in the stainless-steel reaction chamber. After that, the gas-tight reactor was purged with Ar (99.999%) gas for 30 min to ensure air removal. Subsequently, 50 mL $CO_2$ (99.999%) and 0.2 mL $H_2O$ were injected into the reactor. Monochromatic LED light sources at 365 nm, 420 nm, 450 nm, 520 nm, 590 nm and 620 nm (Perfectlight, Beijing) were used to drive photocatalytic $CO_2$ conversion. The power of the light was measured with a PLS-LED100B photoradiometer (Perfect Light, Beijing, China). The gas products were analysed by a gas chromatograph (Agilent 7890B) equipped with two thermal conductivity detectors using He and Ar as the carrier gases.

**AQE calculations**. The AQEs at different wavelengths were calculated by the following equation. The measured intensities of LED light at 365 nm, 420 nm, 450 nm, 520 nm, 590 nm, and 620 nm are shown in Supplementary Table 1. The reaction area was 9.0 cm$^2$.

$$AQE = \frac{2 \times the\,number\,of\,evolved\,CO\,molecules}{the\,number\,of\,incident\,photons} \times 100\% \qquad (11)$$

**$E_{e,hot}$ and $E_{h,hot}$ calculations**. The $E_{e,hot}$ and $E_{h,hot}$ generated under light excitation by the interband transitions of Au NPs can be calculated from the following two equations[62]:

$$E_{e,hot} = E_F - 1.34\,eV + h\nu \qquad (12)$$

$$E_{h,hot} = E_F - 1.34\,eV \qquad (13)$$

where $E_F$ is the Fermi level of Au sample (–5.1 eV vs. vacuum level)[33], and $h\nu$ represents photon energy, which can be given by $h\nu = 1240/\lambda$. Meanwhile, the relationship between the standard hydrogen electrode ($E_{SHE}$) and the vacuum energy level ($E_{vac}$) is[33]:

$$E_{vac} = -4.44 - E_{SHE}(pH = 7) \qquad (14)$$

**DFT calculations**. Theoretical calculations were carried out using DFT with Vienna ab-initio simulation package[63]. The projector augmented wave method was adopted to describe the electron-ion interactions[64]. The exchange-correlation potentials were expressed by the Perdew–Burke–Ernzerhof functional with generalized gradient approximation[65]. The wave functions at each k-point were expanded with a plane wave basis set and a 400 eV cutoff energy was set. A $1 \times 1 \times 1$ Monkhorst-Pack grid was selected to conduct the integration of the Brillouin zone[66]. The convergence threshold was set to $1.0 \times 10^{-4}$ eV for electronic relaxation and 0.05 eV/Å for force. The Van der Waals (vdW) with vdW-DF approximation and spin polarization was involved in our current study[67]. The transition states were obtained by the climbing image nudged elastic band method[68].

The Au(111) surface was constructed by cutting the bulk Au along (111) direction and 3 × 3 supercell with three layers was selected as the computational model. During the optimization, the atoms in the last layer were fixed to maintain the bulk structure and the other atoms were allowed to fully relax. A vacuum layer of 15 Å was used along the c direction normal to the surface to avoid periodic interactions.

The free energy diagrams of $CO_2RR$ and OER were estimated (15) according to the method presented by Nørskov et al.[69]:

$$\Delta G = \Delta E + \Delta ZPE - T\Delta S \qquad (15)$$

Where $\Delta E$ is the energy difference of reactants and products obtained by DFT calculations; $\Delta ZPE$ and $\Delta S$ are the energy difference of zero-point energy and entropy, which are evaluated from the vibrational frequencies; $T$ is the temperature, here 298.15 K is considered; The free energy of $H_2O$ in bulk water was calculated in the gas phase under a pressure of 0.035 bar (the equilibrium vapor pressure of $H_2O$ at 298 K). The free energy of $O_2$ was obtained from the free energy change of the reaction $O_2 + 2H_2 \rightarrow 2H_2O$ under the standard condition and the value of −4.92 eV was chosen. The free energy of ($H^+ + e^-$), according to a computational hydrogen electrode model proposed by Nørskov et al, was estimated as the energy of 1/2 $H_2$ under standard conditions. The entropies and vibrational frequencies of the molecules in the gas phase were extracted from the NIST Chemistry WebBook (URL webbook.nist.gov, National Institute of Standards and Technology Chemistry WebBook).

## Data availability

Source data that support the findings of this study are provided in this paper, which can also be available from the corresponding author upon reasonable request. Source data are provided in this paper.

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

## Acknowledgements

Y. Li acknowledges the support from the National Natural Science Foundation of China (Grant No. 21972082). Y. Wang thanks the support from the National Natural Science Foundation of China (Grant Nos. 21831003, 21673220), and the Jilin Province Science and Technology Development Program (20200201001JC).

## Author contributions

Y.Li conceived and designed the experiments. W.S. carried out the material synthesis, characterizations, and activity tests. Q.L. participated in material synthesis and char-acterizations. Y.Liu and R.Z. conducted the quasi in situ XPS measurements. W.S. and N.S performed the in situ FT-IR measurements. Y.W. performed DFT calculations. N.S. contributed to improving the quality of the figures. Y.Li. wrote the manuscript. J.Z. and C.W. added to the discussion and assisted with revising the manuscript. Y.Li. and Y.W. supervised the work.

## Competing interests

The authors declare no competing interests.
