## [Peer Review File · Nature Communications]

Molecular-level insight into photocatalytic CO₂ reduction with H₂O over Au nanoparticles by interband transitionsEditorial Note: Parts of this Peer Review File have been redacted as indicated to remove third-party material where no permission to publish could be obtained.

REVIEWER COMMENTS

Reviewer #1 (Remarks to the Author):

This manuscript presents an interesting idea of using light excitation and heat for CO₂ reduction in water. Besides some mechanistic weakness that the manuscript may have, the results are very interesting and deserves being considered in Nature Communications.

- 1) One major point is to highlight better all the previous work from P. Jain's group on very similar ideas, but, as far as I know, those experiments were not done in water. I consider that performing CO₂ photo-reduction in water would be a big step to move forward the field.
- 2) Absorption cross-section of 4 nm Au cluster.
- 3) To me it is not clear from the experiments that there is an electronic effect going on. The linear dependence for the laser power experiment (Fig 2f) may suggest that this is purely a thermal effect. Of course, the light + heat experiment should be compared to a pure thermal one but at much higher temperatures (as the light absorbed by the clusters will end up heating the system as well). This is the weakest part of the manuscript in my opinion.
- 4) The oxide formation fitted in the XPS of fig 4c and fig 4d could be noise or it is not very clear to me. I suggest the authors perform electrochemical characterization of the structures to show the oxide appearance or some other characterization.
- 5) Which is the added value of this catalyst compared to the ones in the table in the SI if the production is lower compared to many semiconductors?
- 6) Literature is not complete. There are many recent references in Plasmonic Catalysis that should be listed. There is also a new book in the topic that highlights the challenges of the field and may fit well for this manuscript.

Reviewer #2 (Remarks to the Author):

Comments:

The authors have reported the good performance of photocatalytic CO₂ reduction to CO with the help of H₂O over quantum sized Au nanoparticles. They have experimentally showed that the photocatalytic activity has been originated from the electron hole pairs generating through interband transitions. In addition, they have performed some experiments along with theoretical calculation to prove that the activity is induced by surface Au-O species formed from H₂O decomposition, that synchronously optimizes the rate determining steps in the CO₂ reduction and H₂O oxidation reactions. However, I still have some doubts on the manuscript that need to be addressed. (Major revision)

1. In some region of TEM image (Fig. 1a) the aggregation of Au nanoparticles has been observed. Please provide more TEM images and explain it properly.
2. The authors should mention about the energy states corresponding to binding energy value like Au 4f_{7/2} in the XPS spectra (Fig. 1d).
3. The authors have mentioned "In addition to absorption from the LSPR effect, a dominant absorption region on the high-energy side is also observed.." But that is not well matched with the spectra. Kindly repeat the analysis and explain it.
4. The authors have mentioned that 520 nm peak is due to LSPR effect whereas interband transition should come in high energy (lower wavelength) region. But, interband transition has been observed upto 600 nm. The authors should clearly explain the interband and intraband region.
5. The authors have discussed that the apparent quantum efficiency (AQE) of the Au NPs for CO production under different light wavelengths, which decreases with increasing wavelength. But from

2b around 500 nm AQE value increased and then again decreased with the increasing wavelength. Explain this anomaly.

6. According to results the CO evolution over Au is caused by CO₂ reduction under contributions from both light and thermal energy. Then why the authors not mention the experiment as photothermal reaction?

7. The calculation of E_{e,hot} and E_{h,hot} should be given in method section.

8. For detection the surface adsorbed species during the photocatalytic process it is better to perform in-situ Fourier Transform Infrared (FT IR) spectroscopy.

9. To prove the active surface Au-O species formed from H₂O decomposition involved in enhancement of activity, performance of in-situ XPS study will be much appreciated.

10. the authors missed very important papers recently published. Please study them.

- "Electronic interaction between transition metal single-atoms and anatase TiO₂ boosts CO₂ photoreduction with H₂O", Energy & Environmental Science (2021) published online DOI:

10.1039/D1EE01574E

- Solar fuels: Research and development strategies to accelerate photocatalytic CO₂ conversion into hydrocarbon fuels", Energy & Environmental Science (2021) published online DOI:

10.1039/d1ee02714j

Reviewer #3 (Remarks to the Author):

This work combines experimental and theoretical tools to investigate the photocatalytic reduction CO₂ to CO reaction on Au nanoparticles with the help of H₂O. The detailed reaction mechanism is proposed referring to the reaction intermediates from experimental observation. Although experimental phenomena are very interesting, I would not recommend for publication in NC and main reasons are as follows.

1 Authors investigated the photocatalytic CO₂ reduction reaction in this work. However, they simulated the reaction processes with electroreduction reaction method. Essentially, there are many approaches for studying photocatalytic processes reported in, such as, Nature Chemistry 2011, 3, 467-472; J. Am. Chem. Soc. 2017, 139, 4390-4398; J. Phys. Chem. Lett. 2021, 12, 1125-1130; ACS Nano 2019, 13, 9944-9957. The reaction mechanism might be different if the photo-excitation effect is incorporated.

2. In line 37-39 of page 3, authors claimed that some plasmonic or nonplasmonic metal nanoparticles have been reported. However, the references 6-8 are not related to the corresponding contents.

3. To explain the reduction reaction of CO₂ and H₂O dissociation, authors draw the energetic scheme of photoinduced carrier in Figure 1f and Figure 2i. In the realistic reactions, carriers should have long enough life times in high energy states for carrier injections to reactants. However, in the energy regimes shown in Figure 1f and 2i, the carrier life times may be about 30 fs because carriers can quickly relax to the regime around Fermi level (see Brown et. al., ACS Nano 2016, 10, 957-966). This behavior of carriers is ignored and the mechanism proposed in this work needs to be further demonstrated.

4. In line 134-137 of page 7, authors claimed that "a linear dependence of the photocatalytic reaction rate on temperature under constant light intensity is shown in Fig. 2g, which is induced by the increased energy levels of the photoinduced electrons and the improved populations of the adsorbate in excited vibrational states at higher temperatures". Essentially, the carrier energy increase caused by temperature change with 200 K is quite small compared to photo-induced energy. The mechanism should be further clarified.

5. In Figure 3a, the peaks of IR have been assigned to the species. The results show that the peak

intensities of intermediates of CO₂⁻, COOH, H₂O always increase from 3h to 9h. Conventionally, the concentration of intermediates should increase at first and then decrease during the reactions. Authors should explain the reaction carefully.

6. The H₂ evolution reaction should be considered because it competes with CO₂ reduction reaction.

Dear Reviewers,

We earnestly appreciate for your warm work related to our manuscript. We considered all the comments and fully addressed them in the revised manuscript. These comments were very helpful to raise the quality of the manuscript. Revised parts are in red in the revised manuscript. Our point-by-point responses are attached below.

Manuscript ID: NCOMMS-21-46872

“Molecular-level insight into photocatalytic CO₂ reduction with H₂O over Au nanoparticles by interband transitions”

Sincerely,

Yingxuan Li, Ph. D.

Professor of School of Environmental Science and Engineering,

Shaanxi University of Science and Technology

Weiyang, Xi'an Shaanxi 710021, P. R. Chin

Response to Reviewer 1's Comments:

This manuscript presents an interesting idea of using light excitation and heat for CO₂ reduction in water. Besides some mechanistic weakness that the manuscript may have, the results are very interesting and deserves being considered in Nature Communications.

Response: We appreciate the positive comments on our manuscript. Based on the comments below, we further improved the quality of the manuscript.

1) One major point is to highlight better all the previous work from P. Jain's group on very similar ideas, but, as far as I know, those experiments were not done in water. I consider that performing CO₂ photo-reduction in water would be a big step to move forward the field.

Response: We deeply appreciate the reviewer's recognition of our work.

2) Absorption cross-section of 4 nm Au cluster.

Response: According to the reviewer's suggestion, the calculated interband absorption cross-section of 4 nm Au NPs as a function of wavelength is given in Fig. R1. The specific calculation method is similar to that previously reported by Youngsoo Kim et al (*Nat. Chem.* 2018, 10, 763–769). As shown in Fig. R1, the interband absorption cross-section increases at shorter wavelengths, which is in agreement with the AQE trend in Fig. 2b.

Fig. R1. Calculated wavelength-dependent interband absorption cross-section of 4 nm Au NPs (Supplementary Fig. 2 in the revised manuscript).

Action: We added the calculated wavelength-dependent interband absorption cross section of 4 nm Au NPs and the related discussion to the revised manuscript (on **Page 7**). Furthermore, the method used to calculate the absorption cross section was added to the **Methods section** of the revised manuscript (on **Page 19**).

“Furthermore, the absorption cross-section of the Au NPs in the range of 320–485 nm increases at shorter wavelengths (Supplementary Fig. 2), which is in agreement with the AQE trend in Fig. 2b. This result further confirms that the photocatalytic CO₂RR activity is correlated with the 5d–6sp interband transitions of Au³³.”

Methods: *The absorption cross-section of 4 nm Au NPs was calculated using the method in a previous report³³.*

3) To me it is not clear from the experiments that there is an electronic effect going on. The linear dependence for the laser power experiment (Fig 2f) may suggest that this is purely a thermal effect. Of course, the light + heat experiment should be compared to a pure thermal one but at much higher temperatures (as the light absorbed by the clusters will end up heating the system as well). This is the weakest part of the manuscript in my opinion.

Response: We thank the reviewer for the comment. To the best of our knowledge, the thermal conversion of CO₂ on Au can only be achieved by using H₂ as a reducing agent, which can take place at approximately 200 °C (*Nat. Commun.* 2017, 8, 14542). However, in our study, CO₂-to-CO conversion was achieved with H₂O over Au nanoparticles. Note that thermal splitting of CO₂ or H₂O usually requires high temperatures above 1000 °C, and only metal oxide (MO_x)-based processes have been reported (*Science*, 2010, 330, 1797–1801; *Energy Environ. Sci.*, 2016, 9, 2400–2409; *Int. J. Hydrogen Energy* 2019, 44, 34–60). This thermochemical cycle process can be schematically defined as follows:

In this cycle process, the metal oxide is first reduced at low oxygen partial pressures and elevated temperatures (above 1400 °C). Following thermal reduction, the oxygen-deficient metal oxide is reoxidized at temperatures of ~ 1000 °C with CO₂ and/or H₂O, thereby yielding CO and/or H₂. Due to the stability of CO₂ and H₂O, achieving CO₂ and H₂O splitting on Au at 200 °C through only thermal input is almost impossible. Therefore, it is reasonable to speculate that the CO₂-to-CO conversion in the present study should be dominated by photoinduced electrons.

The reviewer's question, "Of course, the light + heat experiment should be compared to a pure thermal one but at much higher temperatures (as the light absorbed by the clusters will end up heating the system as well)", can be addressed based on the following consideration. As we know that the photothermal mechanism on plasmonic photocatalysts works only when the intensity of the incident light is significant large (in the range of $1-10^6 \text{ W cm}^{-2}$) (*Nat. Mater.* 2015, 14, 567–576; *Catal. Sci. Technol.*, 2018, 8, 3718–3727; *Nat. Commun.* 2020, 11, 5149; *Nat Energy* 2021, 6, 807–814). However, the photothermal effect on catalytic CO_2 reduction can be neglected because 420 nm LED light with very low intensity (73 mW cm^{-1} , less than solar intensity) was used in our work (Table S1 in the revised manuscript). For example, under 420 nm LED light irradiation, the temperature of the sample can only increase from ambient temperature ($24 \text{ }^\circ\text{C}$) to $33 \text{ }^\circ\text{C}$. Therefore, the photothermal effect of the incident light on CO_2 reduction can be easily excluded.

Action: An explanation of the impossibility of the thermal catalytic conversion of CO_2 with H_2O on Au and the discussion stating that the photothermal effect has little effect on the current CO_2 reaction were included in the revised manuscript (on **Page 8**).

"As we know, thermal splitting of CO_2 or H_2O is only reportedly achieved on metal oxide-based catalysts by using oxygen vacancies as the reaction intermediate, which usually requires significantly high temperatures above $1000 \text{ }^\circ\text{C}$ ⁵. Although thermal conversion of CO_2 on Au can be achieved at approximately $200 \text{ }^\circ\text{C}$, H_2 must be used as a reducing agent in this reaction⁵. Therefore, thermocatalytic reduction of CO_2 with H_2O can hardly be achieved on Au at $200 \text{ }^\circ\text{C}$ (Fig. 2c). Furthermore, the photothermal effect on catalytic CO_2 reduction on Au NPs can be neglected because 420 nm LED light with low intensity (73 mW cm^{-1} , even lower than the solar intensity) was used in the present study. Under 420 nm LED light irradiation, the temperature of the sample can only increase from ambient temperature ($24 \text{ }^\circ\text{C}$) to $33 \text{ }^\circ\text{C}$. Based on the above considerations, we can speculate that CO_2 -to- CO conversion on Au NPs should be reduced by the photogenerated electrons."

4) The oxide formation fitted in the XPS of fig 4c and fig 4d could be noise or it is not very clear to me. I suggest the authors perform electrochemical characterization of the structures to show the oxide appearance or some other characterization.

Response: We thank the reviewer for the comment. According to the reviewer's suggestion, electrochemical linear scan voltammetry (LSV) measurements were carried out to prove the formation of Au oxide species during the photocatalytic process. As shown in Fig. R2a, unlike the fresh Au sample, a peak at approximately 0.8 V (vs. RHE) can be clearly observed for the

Au sample after the reaction, and it can be assigned to the reduction peak of Au oxide species (*J. Am. Chem. Soc.* 2007, 129, 42–43). As shown by the cyclic voltammetry (CV) curve in Fig. R2b, the onset potential during the back scan is not sufficient to oxidize Au. The results of Fig. R2a and Fig. R2b prove that oxide species formed on the surface of Au nanoparticles during the photocatalytic CO₂RR.

Furthermore, we used Raman spectroscopy to further characterize the Au–O species formed on the Au surface. As shown in Fig. R2c, a characteristic peak at 480 cm⁻¹ appeared during the photocatalytic reaction process, which can be defined as the stretching vibrations of Au–O species (*ACS Catal.* 2020, 10, 12716–12726). Both the electrochemistry and Raman characterizations confirm the formation of Au–O species, which is consistent with the XPS results in Fig. 4d.

Action: To further demonstrate the formation of Au–O species on the Au surface, electrochemistry and Raman characterizations were carried out in the revised manuscript. We added the new data and corresponding discussions of these characterizations to the revised manuscript (on **Pages 14–15**). The details for performing these tests were also added to the **Methods section** of the revised manuscript (on **Pages 19–20**).

Fig. R2. **a** LSV curves of fresh and used (3 h reaction) Au catalysts (Fig. 4e in the revised manuscript). **b** CV curve of fresh Au (Supplementary Fig. 9 in the revised manuscript). **c** Raman spectra of Au NPs at photocatalytic reaction times of 0 (fresh), 1, 2, and 3 h (Fig. 4f in the revised manuscript).

“To further confirm the formation of Au oxide species, electrochemical linear scan voltammetry (LSV) was performed on the Au samples before and after the photocatalytic reaction. As shown in Fig. 4e, compared to the fresh Au sample, a peak at approximately 0.8 V vs. the reversible hydrogen electrode (RHE) can be clearly observed for the Au sample after 3 h of photocatalytic reaction. As proven by the cyclic voltammetry (CV) curve of Au NPs (Supplementary Fig. 9),

the onset potential during the back scan is not sufficient to oxidize Au. Therefore, the peak at 0.8 V can be attributed to the reduction of the generated Au oxide species during the photocatalytic process⁵¹. In addition, Raman spectroscopy was also used to characterize the Au–O species formed on the Au surface. As shown in Fig. 4f, a characteristic peak at 480 cm⁻¹ appears during the photocatalytic CO₂RR process, which can be attributed to the stretching vibrations of Au–O species⁵². Both the electrochemistry and Raman characterizations confirm that Au–O species formed on the surface of Au NPs during the photocatalytic CO₂RR, which is consistent with the XPS result in Fig. 4d.”

Methods: *Electrochemical measurements were performed in 0.1 M H₂SO₄ electrolyte at room temperature with an electrochemical workstation (CHI 760E, CH Instruments, Inc.) with a standard three-electrode system. The prepared sample, Pt wire, and a saturated Ag/AgCl electrode were used as working, counter, and reference electrodes, respectively. The working electrode was prepared by attaching Au samples onto a 1 × 1 cm fluorine-doped tin oxide glass substrate through dilute Nafion solution (1%). CV and LSV curves were acquired from 0.6 V to 0 V vs. Ag/AgCl at a scan rate of 20 mV/s. The potentials vs. Ag/AgCl were converted to the reversible hydrogen electrode (RHE) scale by the following equation:*

$$E_{RHE} = E_{Ag/AgCl} + 0.197 + 0.059 \times pH \quad (10)$$

Raman microscopy was performed using the emission of a 633 nm laser. The TRPL spectrum was obtained on an Edinburgh FS-5 spectrometer with an excitation wavelength of 340 nm.

5) Which is the added value of this catalyst compared to the ones in the table in the SI if the production is lower compared to many semiconductors?

Response: The reviewer raised a good question. Semiconductor-based photocatalysts have been well studied and provide an attractive approach for room temperature reactions, but a negative dependence of photocatalytic rates on operating temperature is commonly observed in semiconductor-based photocatalysis because of the relatively low Debye temperatures of the semiconductors involved and significant recombination of photogenerated electron–hole pairs, even at moderately elevated temperatures (*J. Phys. Chem. C* 2018, 122, 15, 8045–8057). Furthermore, previous reports have also indicated that the reaction rates for semiconductor photocatalysts depend on the square root of the light intensity (*Nat. Commun.* 2017, 8, 14542). Therefore, in practical applications, if we want to increase the yield of products, the efficiencies for semiconductor photocatalysts can be significantly reduced by increasing the light intensity, which might be a potential factor that can hinder the use of semiconductor photocatalysts because high-intensity light, especially IR light, will inevitably increase the temperature of the

gas–solid reaction system.

Unlike semiconductor-based photocatalysts, metal nanostructures such as the Au NPs in our work can efficiently couple light and thermal energy to drive chemical transformations. The linear growth of photocatalytic reaction rates with increasing temperature and light intensity (Fig. 2d and 2e) suggests that Au photocatalysts with unique capacity represent a class of promising materials that can drive catalytic transformations (*Nat. Mater.* 2012, 11, 1044–1050).

Furthermore, according to the recent results in Supplementary Table 2, the reaction rates of photocatalytic CO₂ conversion processes using H₂O as the hydrogen source are on the order of $\mu\text{mol g}^{-1} \text{h}^{-1}$. However, CO₂ reduction with H₂O on Au in the present study exhibited a CO production rate of $4.2 \text{ mmol g}^{-1} \text{h}^{-1}$, which is in sharply contrast to that of semiconductor photocatalysts.

Action: To make this point clear, we added the following discussion to the revised manuscript (on **Page 9**). The superiority of the present Au photocatalysts for CO₂ RR can be found in Supplementary Table 2, which has been discussed in our original manuscript (at the end of **Page 6** in the revised manuscript).

“The results in Fig. 2d and Fig. 2e make the Au NPs practical for increasing the reaction rate by increasing the temperature and light intensity, indicating that the Au NPs in our work can efficiently couple thermal and light energy to drive chemical transformations, which is in sharp contrast to semiconductor photocatalysts²⁵.”

6) Literature is not complete. There are many recent references in Plasmonic Catalysis that should be listed. There is also a new book in the topic that highlights the challenges of the field and may fit well for this manuscript.

Response: We thank the reviewer for indicating this problem. We added recent literature on plasmonic catalysis to the revised manuscript. In addition, we are also grateful to the reviewer for providing this excellent book (*ISBN: 978-3-527-34750-6 Plasmonic Catalysis: From Fundamentals to Applications Edited by Pedro H.C. Camargo & Emiliano Cortés*). This book enabled us to have a deeper and comprehensive understanding of plasmonic catalysis, and it will be of great help to research in our follow-up work.

Action: We added the corresponding literature to the revised manuscript (on **Page 3**).

“Other than semiconductor photocatalysts, some plasmonic (such as Ag, Au, and Cu)⁶⁻¹² or nonplasmonic (such as Pd and Pt)^{13,14} metal nanoparticles (NPs) have been recently reported to show photocatalytic activities for CO₂ reduction.”

Response to Reviewer 2's Comments:

The authors have reported the good performance of photocatalytic CO₂ reduction to CO with the help of H₂O over quantum sized Au nanoparticles. They have experimentally showed that the photocatalytic activity has been originated from the electron hole pairs generating through interband transitions. In addition, they have performed some experiments along with theoretical calculation to prove that the activity is induced by surface Au-O species formed from H₂O decomposition, that synchronously optimizes the rate determining steps in the CO₂ reduction and H₂O oxidation reactions. However, I still have some doubts on the manuscript that need to be addressed. (Major revision)

Response: We appreciate the positive comments on our manuscript. Based on the comments below, we further improved the quality of the manuscript.

1. In some region of TEM image (Fig. 1a) the aggregation of Au nanoparticles has been observed. Please provide more TEM images and explain it properly.

Response: We agree with the reviewer on this comment. According to the reviewer's suggestion, we performed TEM characterization of Au NPs again (Fig. R3). As shown in Fig. R3a, most of the Au NPs with quasispherical morphology are uniformly dispersed on the carbon film. As shown in the enlarged image (Fig. R3b), accidental aggregation of the Au NPs is also observed, which is understandable considering that the surfactant-free synthesis strategy was used in the present study. Correspondingly, the particle size distribution of Au NPs was remeasured (Fig. R4).

Fig. R3. TEM images of Au NPs: **a** low-magnification TEM image (Fig. 1a in the revised manuscript) and **b** high-magnification TEM image (Supplementary Fig. 1 in the revised manuscript).

Action: We replaced Fig. 1a and 1b in the revised manuscript with the new TEM image (Fig. R3a) and the particle size distribution (Fig. R4), respectively (on **Page 35**). An enlarged image of the TEM image (Fig. R3b) is shown in Supplementary Fig. 1. The following discussion was

added to the revised manuscript (on **Page 5**).

Fig. R4. Particle size distribution of Au NPs (Fig. 1c in the revised manuscript).

“The representative transmission electron microscopy (TEM) image in Fig. 1a shows that the as-prepared Au NPs exhibit quasispherical morphology. As shown in the enlarged image (Supplementary Fig. 1), accidental aggregation of the Au NPs is observed, which is understandable considering that the surfactant-free synthesis strategy was used in the present study.”

2. The authors should mention about the energy states corresponding to binding energy value like Au 4f_{7/2} in the XPS spectra (Fig. 1d).

Response: We thank the reviewer for the suggestion. The binding energies at 83.6 and 87.2 eV corresponding to two spin-orbit splits Au 4f_{7/2} and Au 4f_{5/2} were added to the XPS spectrum of Au NPs (Fig. R5).

Action: According to the reviewer’s suggestion, we added two spin-orbit splits, Au 4f_{7/2} and Au 4f_{5/2}, to the XPS spectra in Fig.1d. (on **Page 35**).

Fig. R5. Au 4f XPS spectra of as-prepared Au NPs (Fig. 1d in the revised manuscript).

3. The authors have mentioned “In addition to absorption from the LSPR effect, a dominant absorption region on the high-energy side is also observed.” But that is not well matched with the spectra. Kindly repeat the analysis and explain it.

Response: Our description in Fig. 1e is indeed not well matched with the spectrum. For example, the absorption edge of Au NPs was not 600 nm, as described in our original manuscript. We thank the reviewer for raising this question, which avoids misleading the readers. In fact, the Au NPs show an absorption band with a tail that can reach 800 nm (Fig. R6a), which is determined by the interband transitions from the 5d band to unoccupied states in the 6sp band above E_F (occurring in the vicinity of the X-point in the Brillouin zone, Fig. R6b). This observation is consistent with the theoretical calculation of the absorption of Au NPs (*Phy. Rev. B* 2003, 68, 115433). Furthermore, as shown in Fig. R6a, a much stronger absorption in the range of 320–516 nm is also attributed to the interband transitions, which is caused by the transition from the top of the 5d band to states just above E_F in the 6sp band (occurring in the vicinity of the L-point in the Brillouin zone, Fig. R6b). For more detailed explanation on the absorption spectrum of the Au NPs, please see our reply to question 4 in the following.

Fig. R6. a UV-Vis absorption spectrum of the Au NPs with interband and intraband absorption (Fig. 1e in the revised manuscript). **b** Schematic diagram of interband and intraband (LSPR) transitions (Fig. 1f in the revised manuscript).

Action: The interband transitions of Au NPs were reanalyzed in the revised manuscript, and we also revised the description of the absorption range of the UV-Vis absorption spectrum of Au NPs (on Pages 5-6). Please see our reply to question 4 for more details.

4. The authors have mentioned that 520 nm peak is due to LSPR effect whereas interband transition should come in high energy (lower wavelength) region. But, interband transition has been observed up to 600 nm. The authors should clearly explain the interband and intraband

region.

Response: We agree that our explanation of the interband and intraband regions of Au was not clear in the previously submitted manuscript. We thank the reviewer for indicating this problem. From the UV–Vis absorption spectrum in Fig. R6a, we can clearly observe that a small characteristic peak (with a maximum absorption at 520 nm) from 485 to 650 nm is ascribed to the LSPR effect due to intraband absorption. The intraband transitions occur by the excitation of free conduction electrons near the Fermi surface from $6sp$ -hybridized atomic orbitals ($6sp$ band), which are energetically allowed, even with low-energy photons in the visible region, and increase until they approach the surface plasmon frequency.

The calculated schematic diagram of interband transitions is shown in Fig. R7 (Right), which has been reported by Beversluis et al (*Phy. Rev. B* 2003, 68, 115433). It can occur at the L-point in the Brillouin zone with a threshold energy of 2.4 eV. This means that at least 2.4 eV (light wavelength $\lambda < 517$ nm) of the photon energy is required for the excitation of electrons from the Au $5d$ band to states just above the Fermi level (E_F) in the $6sp$ conduction band to overcome the interband transition threshold. Moreover, a tail of the interband threshold can extend to 1.8 eV at the X-point in the Brillouin zone (Fig. R7 (Left)), which occurs by the excitation of electrons from the $5d$ band to unoccupied states in the $6sp$ band above E_F with the threshold energy $E_X = 1.8$ eV (light wavelength $\lambda < 689$ nm). Considering that minor discrepancies between the theoretical prediction and experimental observation should be acceptable, the intraband and interband absorption regions of Au are marked with different colours in Fig. R6a

[redacted]

Action: In the revised manuscript, we have clearly explained the interband and intraband absorption region based on the theoretical calculation and experimental results in the revised manuscript. We have modified Fig. 1e and f to express the interband and intraband absorption processes more clearly. To better clarify this point, the corresponding discussions were added into the revised manuscript (on **Pages 5-6**).

“As revealed by the UV–Vis absorption spectrum in Fig. 1e, the Au NPs exhibit absorption in the wavelength range of 320–800 nm. A small absorption peak from 485 to 650 nm with

maximum absorption at 520 nm is ascribed to the LSPR effect due to intraband absorption, which occurs by the excitation of free conduction electrons near the Fermi surface from 6sp-hybridized atomic orbitals of Au (Fig. 1f). In addition to absorption from the LSPR effect, a dominant absorption band with a tail that can reach 800 nm is observed in Fig. 1e, which can be attributed to the 5d–6sp interband transitions of Au NPs (Fig. 1f)^{26,30}. According to theoretical calculations³¹, the interband absorption of Au NPs is derived from two types of excitations. As shown in Fig. 1f, one type of excitation occurs by the transition of electrons from the top of the 5d band to states just above E_F (Fermi level) in the 6sp band with a threshold at 2.4 eV ($\lambda < 517$ nm, occurs near the L-point in the Brillouin zone), and the other is from the 5d band to unoccupied states in the 6sp band above E_F with a tail at 1.8 eV ($\lambda < 689$ nm, occurs near the X-point in the Brillouin zone). Considering that minor discrepancies between the theoretical prediction and experimental observation should be acceptable, the intraband and interband absorption regions of Au are marked with different colours in Fig. 1e. As shown in Fig. 1e, the absorption of Au NPs in the range of 320–800 nm is attributed to the overlap of the LSPR intraband absorption and the interband absorption.”

5. The authors have discussed that the apparent quantum efficiency (AQE) of the Au NPs for CO production under different light wavelengths, which decreases with increasing wavelength. But from 2b around 500 nm AQE value increased and then again decreased with the increasing wavelength. Explain this anomaly.

Response: We thank the reviewer for the comment. As shown in Fig. 2b, the AQE values decrease with increasing wavelength in the wavelength range from 365 to 620 nm, except for a slight increase at 520 nm. This enhanced AQE at approximately 520 nm should be due to the improved light absorption via the typical LSPR effect of the Au NPs.

Action: In the revised manuscript, we indicated that a slight increase in AQE was observed at 520 nm. Moreover, the reasons for this phenomenon are also given in the revised manuscript. To better clarify this point, the corresponding discussions were added to the revised manuscript (on **Page 7**).

“Fig. 2b shows the apparent quantum efficiency (AQE) of the Au NPs for CO production under light with different wavelengths, which decreases with increasing wavelength in the range of 365 to 620 nm, except for a slight increase at 520 nm. The enhanced AQE at 520 nm should be ascribed to the improved light absorption due to the LSPR effect of the Au NPs³².”

6. According to results the CO evolution over Au is caused by CO₂ reduction under

contributions from both light and thermal energy. Then why the authors not mention the experiment as photothermal reaction?

Response: We thank the reviewer for raising this good question. Photothermal catalysis draws on both the thermal and photochemical contributions of light, in which the temperature rise of the reaction system is caused by light illumination (*Energy Environ. Sci.*, 2019, 12, 1122–1142). Therefore, the photothermal reaction is usually achieved at high light intensity in the range of $1\text{--}10^6\text{ W cm}^{-2}$ (*Catal. Sci. Technol.*, 2018, 8, 3718–3727; *Nat. Commun.* 2020, 11, 5149; *Nat Energy* 2021, 6, 807–814). However, low-intensity LED light (420 nm) was used as the light source in the current work, and the maximum light intensity of the 420 nm LED light was only 73 mW cm^{-2} (even less than the solar intensity). Under 420 nm LED light irradiation, the temperature of the sample can only increase from ambient temperature ($24\text{ }^\circ\text{C}$) to $33\text{ }^\circ\text{C}$, indicating that the light-induced thermal effect can be neglected during the reaction process. In our study, an external heating element was used to control the reaction temperature of photocatalytic CO_2 reduction, which is different from photothermal catalysis at this point. Therefore, the reactions in our study were not regarded as a photothermal process.

Action: To clarify this point, we added a discussion to the revised manuscript (on **Page 8**).

“Furthermore, the photothermal effect on catalytic CO_2 reduction on Au NPs can be neglected because 420 nm LED light with low intensity (73 mW cm^{-1} , even lower than the solar intensity) was used in the present study. Under 420 nm LED light irradiation, the temperature of the sample can only increase from ambient temperature ($24\text{ }^\circ\text{C}$) to $33\text{ }^\circ\text{C}$. Based on the above considerations, we can speculate that CO_2 -to- CO conversion on Au NPs should be reduced by the photogenerated electrons.”

7. The calculation of $E_{e,\text{hot}}$ and $E_{h,\text{hot}}$ should be given in method section.

Response: We agree with the reviewer that the calculation of $E_{e,\text{hot}}$ and $E_{h,\text{hot}}$ should be given in the Methods section.

Action: According to the reviewer’s suggestion, we moved the calculation parts of $E_{e,\text{hot}}$ and $E_{h,\text{hot}}$ to the **Methods section**. Please see the details in the revised manuscript (on **Page 22**).

8. For detection the surface adsorbed species during the photocatalytic process it is better to perform in-situ Fourier Transform Infrared (FT IR) spectroscopy.

Response: We thank the reviewer for the suggestion. We agree that it is better to perform *in situ* Fourier transform infrared (*in situ* FT-IR) spectroscopy, which is helpful for providing a

deep understanding of the CO₂ reduction mechanism on Au NPs.

Action: According to the reviewer's suggestion, we conducted *in situ* FT-IR measurements to explore the surface adsorbed species during the photocatalytic process (Fig. R8). This test is helpful for providing insight into the surface chemistry responsible for CO₂ reduction. The instrument used for *in situ* FT-IR measurements is shown in Fig. R9. As shown in Fig. R8, the corresponding characteristic peaks of the *in situ* FT-IR analysis are consistent with the FT-IR analysis results based on the off-line method in our original manuscript. The discussions on the FT-IR spectra in our original manuscript were replaced by those based on the *in situ* FT-IR measurement in the revised manuscript (on **Pages 10–11**). Moreover, the IR absorption intensity changes of reactants (CO₂ and H₂O coadsorbed on Au) and intermediate species (*COOH on Au) generated during the photocatalytic reaction are also shown in Fig. R10, and the related discussion was added to the manuscript (on **Pages 10–11**). The details for performing *in situ* FT-IR measurements were also added to the **Methods section** of the revised manuscript (please see more details on **Page 20** of the revised manuscript).

*“To elucidate the catalytic mechanism, in situ Fourier transform infrared (in situ FT-IR) spectroscopy was conducted to detect the surface-adsorbed species on Au during the photocatalytic process. As shown in Fig. 3a, a broad peak centred at 1650 cm⁻¹ corresponding to the surface-adsorbed H₂O appears with prolonged reaction time, suggesting that H₂O was adsorbed on the Au NPs. As the reaction proceeded, a new peak at 1698 cm⁻¹ that can be assigned to the vibration of carboxylate (CO₂⁻) was observed, which represents the chemical adsorption state of CO₂ on Au⁴⁰. The interaction between the CO₂ and H₂O (and/or surface oxygen atom) coadsorbed on the Au surface leads to the monodentate carbonate (m-CO₃²⁻) peaks at 1510 cm⁻¹ and 1380 cm⁻¹ and bidentate bicarbonate (b-HCO₃²⁻) peak at 1458 cm⁻¹^{40,41}. Moreover, two new peaks at 1545 and 1180 cm⁻¹ that can be ascribed to the *COOH group (the asterisk denotes the surface adsorption site)⁴⁰ appear with increasing reaction time, demonstrating that *COOH is a key intermediate of the CO₂-to-CO conversion process. Moreover, the changes in the peak intensities of the critical intermediates with reaction time are shown in Fig. 3b, which increase with the extension of the reaction time from 0 to 15 min. After that, the peak intensities are slightly reduced with the reaction time being prolonged from 15 to 30 min and are nearly stable after 30 min of reaction. The changes in peak intensities may be understood by considering the cooperative effect between the H₂O and CO₂ adsorption process and the photocatalytic conversion process of the intermediate species during the reduction of CO₂ on Au NPs.”*

Fig. R8. *In situ* FT-IR spectra of the photocatalytic CO₂ reduction process occurring with H₂O on Au. FT-IR spectra were recorded every 5 min (Fig. 3a in the revised manuscript).

Fig. R9. Photo of the instrument used for *in situ* FT-IR measurements (Supplementary Fig. 11 in the revised manuscript).

Fig. R10. Absorbance of the intermediate species that formed on Au during the photocatalytic CO₂RR (Fig. 3b in the revised manuscript).

Methods: *In situ* FT-IR measurements were carried out on a Bruker Vertex 70v spectrometer equipped with a liquid nitrogen-cooled HgCdTe detector. All spectra were obtained with a resolution of 2 cm⁻¹ and an accumulation time of 32 s. A photo of the instrument used for *in situ* FT-IR measurements is shown in Supplementary Fig. 11. Prior to the measurement, a

vacuum pump was used to evacuate all the gases from the reaction chamber for 1 h. Then, CO₂ and H₂O vapour, as reaction gases, were introduced into the reaction chamber through the gas valve. The temperature of the reaction cell was raised to 200 °C with a heating rate of 5 °C·min⁻¹. After that, the background spectrum in the presence of the sample was collected. Next, the FT-IR spectra were recorded as a function of time, in which 420 nm LED light was used to illuminate the sample by a top quartz window.

9. To prove the active surface Au-O species formed from H₂O decomposition involved in enhancement of activity, performance of in-situ XPS study will be much appreciated.

Response: We agree with the reviewer that the *in situ* XPS study provides solid evidence for confirming the formation of the active Au–O species. In the revised manuscript, quasi *in situ* XPS measurements were performed. In these quasi *in situ* measurements, the chamber used for the photocatalytic CO₂ reaction was separated from the XPS analysis chamber. After the photocatalytic CO₂RR, the Au catalyst in the reaction chamber can be transferred to the analysis chamber through the sample transfer axis for XPS measurements without exposing the sample to air, which ensures that this quasi *in situ* XPS study is similar to the *in situ* process. As shown in Fig. R11, upon irradiation at 200 °C, the peaks corresponding to the oxide state of Au emerged at 84.8 eV and 88.4 eV, and their intensities increase with reaction time, which demonstrates that the Au–O species formed by H₂O decomposition on Au during the reaction process. This is consistent with the analysis results based on the off-line method in our original manuscript. Thank you for this suggestion, which helped us provide solid evidence for the formation of Au–O species on the Au surface.

Action: According to the reviewer's suggestion, we conducted quasi *in situ* XPS measurements to confirm the formation process of the Au–O species on the Au surface during the photocatalytic reaction. The XPS spectra of Au 4f in our original manuscript were replaced with the quasi *in situ* XPS measurement in the revised manuscript (on **Page 38**). The results and discussions of the XPS spectra in our original manuscript were replaced by those based on the quasi *in situ* XPS measurement in the revised manuscript (on **Page 14**). The details for performing *in situ* XPS measurements were also added to the **Methods section** of the revised manuscript (please see more details on **Pages 20–21** in the revised manuscript).

Fig. R11. Quasi *in situ* XPS spectra of Au 4f for photocatalytic CO₂ reduction with H₂O on Au (Fig. 4d in the revised manuscript).

“To study the species formed on Au during the photocatalytic process, quasi in situ XPS measurements were carried out. As shown in Fig. 4d, prior to irradiation, the high-resolution XPS spectrum of Au 4f exhibits two peaks at 83.3 eV and 86.9 eV, which represent the Au 4f_{7/2} and Au 4f_{5/2} states of metallic Au (Au⁰)⁵⁰, respectively. Upon irradiation at 200 °C, the peaks corresponding to the oxide state of Au (Au⁺) emerged at 84.8 eV and 88.4 eV⁵⁰, which demonstrates that the Au–O species formed by H₂O decomposition on Au during the reaction process. As shown in Fig. 4d, the concentration of the Au–O species gradually increased as the reaction time was prolonged from 1 h to 3 h, which coincides with the production of Au–O species in the photocatalytic CO₂RR on Au.”

Method: *Quasi in situ XPS spectra were acquired with a customized SPECS UHV system equipped with a SPECS PHOIBOS 150 analyser, a SPECS XR50 X-ray source, and a SPECS ID DLD detector. The basic pressure of the analysis chamber was less than 3×10^{-10} mbar. The Au catalyst used for the XPS measurement was pressed onto stainless steel mesh. The CO₂ photoreduction process was conducted in a load-lock chamber. After loading the catalyst, the reaction chamber was first treated with 5 mbar CO₂ and H₂O vapour and then heated to 200 °C for the photocatalytic reaction. In this process, 420 nm LED light was used to illuminate the sample. The distance between the light source and the sample was approximately 13.5 cm. After the photocatalytic reactions proceeded for different reaction times (1 h, 2 h, or 3 h), the load-lock chamber was pumped down to high vacuum ($<10^{-9}$ mbar). Then, the sample was transferred to the analysis chamber through the sample transfer axis for XPS measurements without exposing the sample to air.*

10. The authors missed very important papers recently published. Please study them.

- "Electronic interaction between transition metal single-atoms and anatase TiO₂ boosts CO₂ photoreduction with H₂O", Energy & Environmental Science (2021) published online DOI: 10.1039/D1EE01574E

- Solar fuels: Research and development strategies to accelerate photocatalytic CO₂ conversion into hydrocarbon fuels", Energy & Environmental Science (2021) published online DOI: 10.1039/d1ee02714j

Response: We thank you for the nice references. These references are helpful for understanding the photocatalytic reduction of CO₂ with H₂O. We cited these references in the revised manuscript (on **Page 3**).

“Coupling CO₂ reduction and H₂O oxidation over photocatalysts using solar energy is an appealing strategy to alleviate the greenhouse effect and simultaneously produce value-added chemicals or fuels^{1,2}.”

Response to Reviewer 3's Comments:

This work combines experimental and theoretical tools to investigate the photocatalytic reduction CO₂ to CO reaction on Au nanoparticles with the help of H₂O. The detailed reaction mechanism is proposed referring to the reaction intermediates from experimental observation. Although experimental phenomena are very interesting, I would not recommend for publication in NC and main reasons are as follows.

Response: In this revised manuscript, we considered all the comments of the reviewers and tried our best to revise the manuscript accordingly. In our manuscript, we showed an interesting finding that quantum-sized Au nanoparticles exhibit superiority in photocatalytic CO₂ reduction with abundant H₂O over CO₂ reduction with H₂, which could assist the design of catalytic processes based on metal photocatalysts that are more energy efficient and inexpensive than current processes that generally require strong reductants (such as H₂). In addition to the novel mechanistic insight where the increased activity on Au is induced by surface Au–O species formed from H₂O decomposition, as described in our original manuscript, we investigated the reaction mechanism by *in situ* FT-IR and XPS measurements in the revised manuscript, which strengthened the mechanistic understanding of CO₂ reduction with H₂O on Au. In the revised manuscript, we also obtained the time-resolved fluorescence decay spectrum of Au NPs, indicating that electron–hole pairs produced on the quantum-sized Au nanoparticles by interband transitions have long enough lifetimes for photocatalytic conversion of CO₂ and H₂O into CO and O₂. This observation is in sharp contrast with traditional plasmonic excitation on metal NPs, which shows the novelty of our manuscript. Finally, to avoid misleading theoretical calculations, we mainly focus on explaining the role of surface Au–O species in CO₂ reduction reactions from a thermodynamic point of view, which does not affect the original conclusions of our manuscript. Our findings could open a promising new route for directly using metal nanoparticles as photocatalysts for solar energy conversion. In short, the modifications in the revised manuscript enhanced the mechanistic aspects of our paper and further improved and strengthened the present manuscript. We truly appreciate the reviewer to keep in mind these important points while reevaluating this work for Nature Communications.

1. Authors investigated the photocatalytic CO₂ reduction reaction in this work. However, they simulated the reaction processes with electroreduction reaction method. Essentially, there are many approaches for studying photocatalytic processes reported in, such as, Nature Chemistry 2011, 3, 467-472; J. Am. Chem. Soc. 2017, 139, 4390–4398; J. Phys. Chem. Lett. 2021, 12, 1125–1130; ACS Nano 2019, 13, 9944–9957. The reaction mechanism might be different if the photo-excitation effect is incorporated.

Response: We thank the reviewer for providing us the photocatalytic references. We agree that the light effect is important for photocatalytic reactions. However, it is well known that the excited state calculation is very expensive and time-consuming and can only deal with small systems. For example, a Au cluster containing only 18 atoms was built to study the photocatalytic CO₂ decomposition mechanism by time-dependent density functional theory (*J. Phys. Chem. Lett.* 2021, 12, 1125–1130, and the embedded correlated wavefunction (ECW) method can only deal with 10–12 atoms (*ACS Nano* 2019, 13, 9944–9957). Moreover, CO₂ reduction with H₂O is much more complicated, which further limits the application of the excited state calculation in these reactions. It is understandable that each calculation method has its limitations. These limitations might lead to the general use of the electroreduction reaction method in photocatalytic CO₂ reduction (such as *ACS Catal.* 2022, 12, 89–100; *J. Am. Chem. Soc.* 2020, 142, 19259–19267; *J. Am. Chem. Soc.* 2019, 141, 423–430; *J. Am. Chem. Soc.* 2021, 143, 6551–6559; *Nat. Commun.* 2019, 10, 2521). In fact, there is a noticeable absence of a consummate calculation method that can perform simulations that are entirely consistent with the actual experiment.

Based on the experimental analysis through *in situ* FTIR, quasi *in situ* XPS, and ¹H ssNMR, we proved that the photocatalytic CO₂ reduction and H₂O oxidation processes over Au NPs undergo pathways in the order CO₂ → *CO₂ → *COOH → *CO → CO and H₂O → *OH → *O → *OOH → O₂, respectively, which are the same as the electrochemical reduction processes of CO₂ in H₂O (*Adv. Funct. Mater.* 2020, 30, 2003438; *Proc. Natl Acad. Sci. USA* 2019, 116, 23915–23922). To our knowledge, the thermodynamic properties of the CO₂RR on Au should not change by light stimulation. Therefore, the Gibbs free energy diagram (Fig. 5c and 5d) estimated by the electroreduction reaction method is suitable to investigate the effect of surface Au–O species on the photocatalytic CO₂RR mechanism on Au. As shown in Fig. 5c, the significant difference for the Au sample with and without Au–O lies in the reaction energy for desorption of CO. The reaction energies for *CO desorption changed from a large positive reaction energy of 0.68 eV to 0.33 eV in the presence of surface Au–O species, indicating that the adsorbed CO can be much more easily eliminated on Au if H₂O is used for CO₂ reduction. Obviously, this result cannot be changed by light irradiation. In our revised manuscript, we focus on studying the role of O/OH (dissociated from H₂O) in the CO₂RR from a thermodynamic point of view, which do not affect the reaction mechanism and conclusion of our original manuscript. Based on the above analysis, we can reasonably speculate that the calculation method we used in our study might be appropriate for studying the improved photocatalytic CO₂ reduction activities on Au with H₂O.

In the references provided by the reviewer, we see that potential energy surfaces of the ground state usually possess extremely high energy barriers, while the crossing of excited states effectively lowers the reaction energy barrier (O₂, N₂, and NH₃ dissociation) and improves the

reaction activity. In our current DFT simulations, we also found that the energy barrier of H₂O decomposition at the ground state is 1.35 eV (see Fig. 5a), which is relatively high and cannot occur at room temperature. However, if the crossing of excited states is considered, the energy barrier will also be significantly reduced according to the literatures (*Nat. Chem.* 2011, 3, 467–472; *J. Am. Chem. Soc.* 2017, 139, 4390–4398; *J. Phys. Chem. Lett.* 2021, 12, 1125–1130; *ACS Nano* 2019, 13, 9944–9957). This further confirms the rationality of our assumption that H₂O will be rapidly decomposed into O or OH under light and then form an O-preadsorbed Au surface. Similarly, the energy barrier of the CO₂ hydrogenation process in Fig. 5b will be effectively reduced according to the above literature. However, there is still a possibility that the relative energy values in Fig. 5b might change if the excited states are considered. To avoid misunderstanding, we deleted Fig. 5b in the revised manuscript, which does not affect the other results or conclusions of our manuscript.

Action: To avoid misleading, we mainly focus on explaining the role of O/OH (dissociated from H₂O) in CO₂ reduction reactions from a thermodynamic point of view, and Fig. 5b in our original manuscript was deleted, which does not affect the conclusions of our manuscript. Furthermore, the possible effect of light irradiation on CO₂ reduction on Au was discussed based on the references provided by the reviewer. Our discussion of the theoretical calculation was reorganized (please see the details in our revised manuscript), and the following sentences were added to the revised manuscript on **Pages 15, 16 and 17**.

“Based on the above experimental results, we employed density functional theory (DFT) calculations based on an electrochemical method to examine the possible CO₂RR mechanisms occurring on the surface of Au. The experimental results above have proven that the photocatalytic CO₂ reduction and H₂O oxidation processes over Au NPs undergo pathways (Equations (1)-(9)) that are the same as the electrochemical reduction pathways of CO₂ in H₂O⁵⁶. Furthermore, the Gibbs free energy diagram estimated by the electrochemical simulation method has been widely used in investigating the mechanism of photocatalytic CO₂RR considering that the thermodynamic properties of the chemical reactions should not be changed by light stimulation^{41,57}. Based on the above considerations, the electrochemical simulation method might be suitable to investigate the photocatalytic CO₂RR mechanism on Au in terms of thermodynamics.”

“However, according to the previous theoretical calculations, the energy barrier for H₂O decomposition would be significantly reduced if the crossing of excited states is considered^{58,59}. Therefore, both the presence of H₂O and light illumination are critical for the CO₂RR on Au, which is consistent with our experimental observations in Fig. 2c.”

*“As shown in Fig. 5b, the reaction energy for *CO desorption is reduced from 0.68 eV to 0.33 eV in the presence of surface Au–O species, indicating that the adsorbed CO can be more easily eliminated on Au if H₂O is used for CO₂ reduction. This is also helpful for enhancing the CO₂-to-CO conversion activity on Au.”*

2. In line 37-39 of page 3, authors claimed that some plasmonic or nonplasmonic metal nanoparticles have been reported. However, the references 6-8 are not related to the corresponding contents.

Response: It is true that the references concerning nonplasmonic metal nanoparticle-based photocatalysts were not cited. We appreciate the reviewer for finding this problem.

Action: We added the relevant references on nonplasmonic metal nanoparticles to the revised version of manuscript. Furthermore, some of the latest reports and books on plasmonic photocatalysis were also added to the references. Please see more details in the revised manuscript (on **Page 3**).

3. To explain the reduction reaction of CO₂ and H₂O dissociation, authors draw the energetic scheme of photoinduced carrier in Figure 1f and Figure 2i. In the realistic reactions, carriers should have long enough life times in high energy states for carrier injections to reactants. However, in the energy regimes shown in Figure 1f and 2i, the carrier life times may be about 30 fs because carriers can quickly relax to the regime around Fermi level (see Brown et. al., ACS Nano 2016, 10, 957–966). This behavior of carriers is ignored and the mechanism proposed in this work needs to be further demonstrated.

Response: We thank the reviewer for this comment, which reminded us to consider the lifetime of the photogenerated charges on Au NPs. As shown in the reference provided by the reviewer, the lifetime of hot carriers produced in plasmonic excitation on Au NPs is only approximately 30 fs (ACS Nano 2016, 10, 957–966). We agree with the reviewer that the hot carriers generated from the LSPR effect of metal photocatalysts always have a much shorter lifetime than those from semiconductors (Nano Energy 2019, 56, 286–293), making CO₂ reduction with H₂O difficult. This is consistent with our experimental observation in Fig. 2b of our manuscript. As shown in Fig. 2b, photocatalytic CO₂-to-CO conversion on Au NPs is mainly attributed to the hot electrons from interband transitions. In contrast, the apparent quantum efficiency is fairly low under irradiation with wavelengths ranging from 485 to 650 nm, which corresponds to the excitation wavelength of the plasmonic resonance. This result proves that the CO evolution

reaction was not primarily driven by the LSPR effect of Au. Therefore, in the present study, we discussed the photocatalytic mechanism of Au NPs based on the charges produced by interband transitions rather than the hot carriers due to the LSPR effect (i.e., intraband transitions). Note that the interband transitions of Au NPs are similar to the excitations in semiconductors.

To gain deep insight into the charge-transfer dynamics, time-resolved photoluminescence (TRPL) measurements were performed on the Au NPs (Fig. R12). Based on Fig. R12, the carrier lifetime of the Au sample was calculated to be 0.2 ns, which is close to that of semiconductors and is long enough to drive photocatalytic reactions (*Appl. Catal., B* 2022, 301, 120802). This lifetime is also much longer than that produced with the LSPR effect (30 fs). The longer charge lifetime could improve the potential for the involvement of photogenerated charge carriers in photocatalytic reactions before they recombine, which leads to enhanced photocatalytic activity (*ACS Catal.* 2021, 11, 650–658). Moreover, the Au NPs are also proven to have a suitable electronic band structure to achieve simultaneous CO₂ reduction and H₂O oxidation by interband transitions (Fig 2i).

Furthermore, the calculated interband absorption cross-section of 4 nm Au NPs as a function of wavelength is given in Fig. R1. As shown in Fig. R1, the interband absorption cross-section increases at shorter wavelengths, which is in agreement with the AQE trend in Fig. 2b. This result further confirms that the photocatalytic CO₂RR activity is correlated with the *5d–6sp* interband transitions of Au (*Nat. Chem.*, 2018, 10, 763–769). Therefore, in the present manuscript, all the experimental results support that the photocatalytic mechanism of Au NPs is based on interband transitions, which is unlike previous reports in which the activities were mainly induced by the LSPR effect of Au NPs due to intraband transitions.

Fig. R12. TRPL decay spectrum of Au NPs (Supplementary Fig. 3 in the revised manuscript).

Action: A time-resolved photoluminescence decay spectrum was used to evaluate the kinetic properties of the photogenerated charge carriers on Au NPs. The photocatalytic mechanism of

Au NPs is discussed based on the interband transitions rather than the LSPR effect of Au NPs. As shown in the following, the discussions of the TRPL decay spectrum and the TRPL method were added to the revised manuscript on **Page 7** and **Page 19**, respectively. For the calculated interband absorption cross-section of 4 nm Au NPs and the corresponding discussion, please see our reply to question 2 of Reviewer 1 for more details.

“To gain insight into the dynamics of the photogenerated carriers, the time-resolved photoluminescence (TRPL) decay spectrum of Au NPs was obtained (Supplementary Fig. 3). The carrier lifetime of the Au sample was calculated to be approximately 0.2 ns, which is close to that of semiconductor photocatalysts and is long enough to drive photocatalytic reactions³⁴. Interestingly, this lifetime is also much longer than that produced with the LSPR effect (approximately 30 fs)²³, which is beneficial for the transfer of photogenerated charge carriers and thereby leads to improved photocatalytic activity on Au NPs. The TRPL result proves the superiority of the charge carriers induced by the interband transition of Au for photocatalysis.”

Method: *The TRPL spectrum was tested with an Edinburgh FS-5 spectrometer with an excitation wavelength of 340 nm.*

4. In line 134-137 of page 7, authors claimed that “a linear dependence of the photocatalytic reaction rate on temperature under constant light intensity is shown in Fig. 2g, which is induced by the increased energy levels of the photoinduced electrons and the improved populations of the adsorbate in excited vibrational states at higher temperatures”. Essentially, the carrier energy increase caused by temperature change with 200 K is quite small compared to photo-induced energy. The mechanism should be further clarified.

Response: We thank the reviewer for indicating this problem. We agree with the reviewer's opinion that the carrier energy increase caused by the temperature change at 200 K is quite small compared to the photoinduced energy. It is generally proven that there is a synergetic effect of photon flux and thermal energy on thermally assisted photocatalytic reactions. Although the effect of operating temperature on metal-based photocatalysts is complicated, there is a general consensus that raising the reaction temperature would lead to an improved relative population of the adsorbate in excited vibrational states (higher occupancy of excited adsorbate vibrational states), which can make the reactive adsorbate require less energy to overcome the activation barrier (*Nat. Mater.* 2012, 11, 1044-1050; *Angew. Chem., Int. Ed.* 2014, 53, 2935–2940). Therefore, the effect of operating temperature on CO production on Au NPs was explained for this reason in the revised manuscript. Since the purpose of this manuscript is to study the effect of surface Au–O species on the photocatalytic CO₂RR, the influence of

reaction temperature on photocatalytic CO₂ conversion on Au needs to be systematically studied in our future work.

Action: To avoid misleading readers, the effect of operating temperature on CO production on Au NPs was explained by considering the improved populations of the adsorbate in excited vibrational states at higher temperatures. We deleted the explanation based on the increasing energy level of the photoinduced electrons to prevent misunderstandings. We modified the original claim and added the following discussion to the revised manuscript (on **Pages 8–9**).

“However, a linear dependence of the photocatalytic CO production rate on the reaction temperature at constant light intensity is shown in Fig. 2d, which indicates that the heat input can play a role in improving the photocatalytic CO₂RR activity of Au. This phenomenon is possibly induced by the improved populations of the adsorbate in excited vibrational states at higher temperatures¹⁷. In the future, the influence of the reaction temperature on photocatalytic CO₂ conversion on Au needs to be systematically studied.”

5. In Figure 3a, the peaks of IR have been assigned to the species. The results show that the peak intensities of intermediates of CO₂⁻, COOH, H₂O always increase from 3h to 9h. Conventionally, the concentration of intermediates should increase at first and then decrease during the reactions. Authors should explain the reaction carefully.

Response: We thank the reviewer for the comment. To more precisely detect the surface adsorbed species, *in situ* FT-IR measurements (Fig. R13) were performed, by which the changes in surface intermediates during the photocatalytic process can be directly explored. The instrument used for *in situ* FT-IR measurements is shown in Fig. R15. As shown in Fig. R13, the corresponding characteristic peaks of the *in situ* FT-IR analysis results are consistent with the FT-IR analysis results based on the off-line method in our original manuscript. Remarkably, as shown in Fig. R14, we found that the peak intensity of intermediates of COOH* first showed an increasing trend with the extension of the reaction time from 0 to 15 min. Then, the peak intensity slightly decreased with prolonging the reaction time to 30 min. This may be understood by considering the cooperative effect between the H₂O and CO₂ adsorption process and the photocatalytic conversion process of the intermediate species during the reduction of CO₂ on Au NPs.

Fig. R13. *In situ* FT-IR spectra of the photocatalytic CO₂ reduction process occurring with H₂O on Au. FT-IR spectra were recorded every 5 min (Fig. 3a in the revised manuscript).

Fig. R14. Absorbance of the intermediate species that formed on Au during the photocatalytic CO₂RR (Fig. 3b in the revised manuscript).

Fig. R15. Photo of the instrument used for *in situ* FT-IR measurements (Supplementary Fig. 11 in the revised manuscript).

Action: We added the *in situ* FT-IR and corresponding discussion to the revised manuscript (on Pages 10-11). Details on performing *in situ* FT-IR measurements were also added to the

Methods section of the revised manuscript (please see more details on **Page 20** of the revised manuscript).

*“To elucidate the catalytic mechanism, in situ Fourier transform infrared (in situ FT-IR) spectroscopy was conducted to detect the surface-adsorbed species on Au during the photocatalytic process. As shown in Fig. 3a, a broad peak centred at 1650 cm^{-1} corresponding to the surface-adsorbed H_2O appears with prolonged reaction time, suggesting that H_2O was adsorbed on the Au NPs. As the reaction proceeded, a new peak at 1698 cm^{-1} that can be assigned to the vibration of carboxylate (CO_2^-) was observed, which represents the chemical adsorption state of CO_2 on Au⁴⁰. The interaction between the CO_2 and H_2O (and/or surface oxygen atom) coadsorbed on the Au surface leads to the monodentate carbonate ($m\text{-CO}_3^{2-}$) peaks at 1510 cm^{-1} and 1380 cm^{-1} and bidentate bicarbonate ($b\text{-HCO}_3^{2-}$) peak at 1458 cm^{-1} ^{40,41}. Moreover, two new peaks at 1545 and 1180 cm^{-1} that can be ascribed to the *COOH group (the asterisk denotes the surface adsorption site)⁴⁰ appear with increasing reaction time, demonstrating that *COOH is a key intermediate of the CO_2 -to- CO conversion process. Moreover, the changes in the peak intensities of the critical intermediates with reaction time are shown in Fig. 3b, which increase with the extension of the reaction time from 0 to 15 min. After that, the peak intensities are slightly reduced with the reaction time being prolonged from 15 to 30 min and are nearly stable after 30 min of reaction. The changes in peak intensities may be understood by considering the cooperative effect between the H_2O and CO_2 adsorption process and the photocatalytic conversion process of the intermediate species during the reduction of CO_2 on Au NPs.*

Method: *In situ FT-IR measurements were carried out on a Bruker Vertex 70v spectrometer equipped with a liquid nitrogen-cooled HgCdTe detector. All spectra were obtained with a resolution of 2 cm^{-1} and an accumulation time of 32 s. A photo of the instrument used for in situ FT-IR measurements is shown in Supplementary Fig. 11. Prior to the measurement, a vacuum pump was used to evacuate all the gases from the reaction chamber for 1 h. Then, CO_2 and H_2O vapour, as reaction gases, were introduced into the reaction chamber through the gas valve. The temperature of the reaction cell was raised to $200\text{ }^\circ\text{C}$ with a heating rate of $5\text{ }^\circ\text{C min}^{-1}$. After that, the background spectrum in the presence of the sample was collected. Next, the FT-IR spectra were recorded as a function of time, in which 420 nm LED light was used to illuminate the sample by a top quartz window.*

6. The H_2 evolution reaction should be considered because it competes with CO_2 reduction reaction.

Response: We thank the reviewer for raising this question. We agree with the reviewer that the selectivity is a fundamental issue for the CO₂RR with H₂O due to the competing H₂ evolution reaction. To achieve high selectivity in the CO₂RR, the hydrogen evolution reaction (HER) should be suppressed. In fact, the presence of H₂ in the gas mixture was tested by gas chromatography in our original study. However, no H₂ production was observed during photocatalytic CO₂ reduction with H₂O (Fig. R16).

Fig. R16. H₂ production on Au NPs under 420 nm LED light irradiation at 200 °C (Supplementary Fig. 5 in the revised manuscript).

Action: According to the reviewer's suggestion, we added the H₂ production result to the revised manuscript. The situation where no H₂ production occurred in the photocatalytic CO₂ reduction with H₂O process is indicated in the revised manuscript. Furthermore, we added discussions on the suppression of the HER based on the stabilization of the *COOH intermediate on the Au NPs and the weak affinity between Au and H. These discussions on the reason for the suppression of the HER were added to the revised manuscript on **Page 10**.

*“It is well known that selectivity is also a fundamental issue for the CO₂RR with H₂O due to the competing H₂ evolution reaction (HER). In the present study, no H₂ production on Au was detected during photocatalytic CO₂ reduction with H₂O (Supplementary Fig. 5). The results in Fig. 3c-d prove that the *COOH intermediate can be strongly adsorbed on the active sites of the Au NPs (discussed below), which is a possible reason for the HER being suppressed since the active sites are occupied by *COOH. On the other hand, the adsorbed *COOH can increase the reaction barrier for the HER on Au NPs³⁸, which can also contribute to the suppression of the HER. In addition, the weak affinity between Au and H might be another factor responsible for suppressing the HER on Au³⁹. For the above three reasons, selective CO₂ reduction with the inhibition of the side HER is achieved on the Au photocatalyst in the present study.”*

Other corrections

1. Except for the above revisions according to the reviewers' comments, we revised Supplementary Table 1, Fig. 2b, and Fig. 2e in the manuscript because the light intensities in Supplementary Table 1 were wrongly calculated (due to the incorrect calculation of the area of incident light) in our original manuscript. These revisions did not affect the results of these figures. Furthermore, these corrections do not affect the other results and the conclusions of the manuscript. We apologize for this.

Supplementary Table 1. Light intensity corresponding to different wavelengths of monochromatic LED light sources.

Light source	Light intensity (mW•cm ⁻²)
365 nm	48
420 nm	73
450 nm	94
520 nm	33
590 nm	23
620 nm	59

Fig. 2b. Wavelength dependent AQEs of the Au photocatalyst.

Fig. 2e. The light intensity dependent CO production rate on Au under 420 nm LED light irradiation.

2. We have read the formatting instructions of Nature Communications carefully and revised our manuscript accordingly. Please see the details in the revised manuscript.

REVIEWERS' COMMENTS

Reviewer #1 (Remarks to the Author):

I think the authors did a very good job in the revisions and that the article is ready for acceptance.

Reviewer #2 (Remarks to the Author):

The authors modified manuscript well according to the reviewers' comments. It's acceptable.

Reviewer #3 (Remarks to the Author):

The authors modified their manuscript according to the comments and questions from the reviewers. It may be publishable, but several questions should be further considered.

1. The Pd and Pt are indeed plasmonic metal nanoparticles although their plasmon effect is not as good as Cu, Ag, and Au. In authors' reply, they added two references in which Pd and Pt are not defined as the nonplasmonic metal nanoparticles, inconsistent with the authors' claim.
2. It is hard to understand using the electrochemical method to explain the photocatalytic experimental results. Although two papers have been cited to refer to their concept, both of references are semiconductors, and not the pure metal catalysis system. In addition, the authors claimed that computational methods can only deal with small atoms. Essentially, periodical calculation approaches, such as the surface structure method used in Nature Chemistry 2011, 3, 467-472, are applicable to the present system. competes with CO₂ reduction reaction.

Response to Reviewer 1's Comments:

I think the authors did a very good job in the revisions and that the article is ready for acceptance.

Response: We appreciate the positive comments on our manuscript. We earnestly appreciate for your warm work related to our manuscript.

Response to Reviewer 2's Comments:

The authors modified manuscript well according to the reviewers' comments. It's acceptable.

Response: The authors thank the reviewer for the positive feedback. We appreciate you sincerely for your warm work related to our manuscript.

Response to Reviewer 3's Comments:

The authors modified their manuscript according to the comments and questions from the reviewers. It may be publishable, but several questions should be further considered.

Response: We deeply appreciate the reviewer's recognition of our work. The authors thank the reviewer for the thorough review.

1. The Pd and Pt are indeed plasmonic metal nanoparticles although their plasmon effect is not as good as Cu, Ag, and Au. In authors' reply, they added two references in which Pd and Pt are not defined as the nonplasmonic metal nanoparticles, inconsistent with the authors' claim.

Response: It is true that Pd and Pt nanoparticles also have plasmonic properties. We appreciate the reviewer for finding this problem. In the revised manuscript, the references concerning Ni and Ru nanoparticles were cited as the nonplasmonic metal nanoparticle-based photocatalysts.

2. It is hard to understand using the electrochemical method to explain the photocatalytic experimental results. Although two papers have been cited to refer to their concept, both of references are semiconductors, and not the pure metal catalysis system. In addition, the authors claimed that computational methods can only deal with small atoms. Essentially, periodical calculation approaches, such as the surface structure method used in Nature Chemistry 2011, 3, 467-472, are applicable to the present system.

Response: We thank the reviewer for this comment. According to the reviewer's suggestion, a reference concerning Ag nanoparticles that used the electrochemical method to explain the photocatalytic CO₂ reduction results were cited in the revised manuscript. This further indicates that the electrochemical simulation method might be suitable for investigating the mechanism of photocatalytic CO₂RR from a thermodynamic point of view.

We have thoroughly read the reference (Nature Chemistry 2011, 3, 467-472) provided by the reviewer. In this reference, a Ag(100) surface (containing 24 atoms) with periodical boundary condition was built to calculate the potential energy surface for ground-state O₂ and excited state O[•]. However, as CO₂ molecule is larger than O₂, in order to avoid the interaction of periodic boxes, a 3×3×3 supercell model containing 36 atoms was built for the present Au photocatalyst, which is a much more expensive model than the above Ag since the computational time increases exponentially with the number of atoms. Furthermore, if the crossing of the excitation states is considered, the periodic calculation approaches are difficult to deal with such a complicated system and further limit the application of the excited state calculation on the Au photocatalyst.

Furthermore, the interaction between photoinduced electrons and the adsorbed O₂ molecules on the Ag catalyst is significantly different from the CO₂RR on our Au catalyst. In O₂ dissociation process, the excited electron from the Ag surface was directly transferred to the adsorbed O₂ and formed O₂[•] species, which lead to the stretching of O-O bond. In our experiments, the excited electrons were transferred to the H⁺ obtained from H₂O followed by the CO₂ + (H⁺ + e⁻) → COOH and COOH + (H⁺ + e⁻) → CO + H₂O reactions. These complicated reactions further limit the application of the excited state calculation in the present study. These reactions generally takes place in photocatalytic CO₂ reduction processes (Angew. Chem. Int. Ed. 2018, 57, 7610–7627) and are exactly same to those assumed in our simulation based on electrochemical method. Therefore, electrochemical method (computational hydrogen electrode concept) is also suitable to explain the photocatalytic experimental results from a thermodynamic point of view. Moreover, we still agree with the reviewer that the excited state calculation is important for photocatalytic reactions from a kinetic point of view. Therefore, we think that the excited state calculation on CO₂RR of Au catalyst needs to be systematically studied in the future work.